# Serotonin signaling regulates actomyosin contractility during morphogenesis in evolutionarily divergent lineages

Sanjay Karki [1], Mehdi Saadaoui [1], Valentin Dunsing[1], Stephen Kerridge[1], Elise Da Silva[1], Jean-Marc Philippe[1], Cédric Maurange [1] & Thomas Lecuit [1,2] ✉

Serotonin is a neurotransmitter that signals through 5-HT receptors to control key functions in the nervous system. Serotonin receptors are also ubiquitously expressed in various organs and have been detected in embryos of different organisms. Potential morphogenetic functions of serotonin signaling have been proposed based on pharmacological studies but a mechanistic understanding is still lacking. Here, we uncover a role of serotonin signaling in axis extension of *Drosophila* embryos by regulating Myosin II (MyoII) activation, cell contractility and cell intercalation. We find that serotonin and serotonin receptors 5HT2A and 5HT2B form a signaling module that quantitatively regulates the amplitude of planar polarized MyoII contractility specified by Toll receptors and the GPCR Cirl. Remarkably, serotonin signaling also regulates actomyosin contractility at cell junctions, cellular flows and epiblast morphogenesis during chicken gastrulation. This phylogenetically conserved mechanical function of serotonin signaling in regulating actomyosin contractility and tissue flow reveals an ancestral role in morphogenesis of multicellular organisms.

The neurotransmitter serotonin is a monoamine produced by the decarboxylation of the amino acid tryptophan. It is evolutionarily conserved in protozoa and most metazoans studied[1–7] and is also present in plants[8,9]. Serotonin functions via serotonin receptors, which are estimated to have evolved around 800 million years ago, reviewed in ref. 10. Seven subtypes of serotonin receptors (5HT1-5HT7) have been identified in vertebrates[11]; most of which are phylogenetically conserved across many species[12,13], reviewed in ref. 14 including insects such as *Drosophila melanogaster* (fruit fly), *Tribolium castaneum* (beetle), *Aedes aegypti* (mosquito)[15]. All the 5HT receptors except 5HT3R which is a gated ion channel receptor, are seven-pass transmembrane G protein-coupled receptors. Serotonin and G protein-coupled serotonin receptors have a wide range of functions in modulating physiological and behavioral processes in animals. In addition, the serotonergic system has non-neuronal functions such as energy balance, pulmonary, cardiac, gastrointestinal, and reproductive function reviewed in ref. 16.

The morphogenetic function of serotonin was first purposed and reported by Buznikov and colleagues in the 1960s, based on their studies on sea urchin embryos, reviewed in refs. 1,7, where inhibition of serotonin signaling leads to the blockage of extension of the archenteron, the embryonic primitive digestive tube. Subsequent work in sea urchin[17,18], chicken[19–21], *Xenopus*[21–25], mouse[26] and *Drosophila*[27–29] reported the potential role of serotonin in driving morphogenesis. In the chicken, serotonin has been detected in the embryo during primitive streak formation[21]. Morphological defects have been characterized during embryogenesis following the pharmacological treatments targeting the serotonin receptors[19–21], however the link to the morphogenetic processes/mechanisms, which were not understood at the time, was lacking.

[1]Aix-Marseille Université & CNRS, IBDM—UMR7288 & Turing Centre for Living Systems, Marseille, France. [2]Collège de France, Paris, France. ✉e-mail: thomas.lecuit@univ-amu.fr

Large-scale directed tissue-flow is observed during morphogenesis in many organisms. For example, during the primitive streak formation in chicken embryos and axis extension (germ-band extension) in *Drosophila,* reviewed in ref. 30. Cell intercalation by polarized remodeling of the cell-cell contacts via actomyosin contractility drives tissue extension. Apical cell-cell contacts contract along the midline and then extend along the anterior-posterior axis resulting in tissue convergence and extension along these axes[31–35] and reviewed in ref. 30. In *Drosophila*, polarized actomyosin contractility drives the polarized junctional remodeling and hence the polarized cell rearrangements[32,33]. Toll receptors (Toll 2, 6 and 8) are required for the planar polarization of MyoII junctional recruitment[36,37]. A recent study showed that the Toll-8 and the adhesion GPCR Cirl/Latrophilin interact and self-organize to form a polarizing field to facilitate asymmetric activation of MyoII at the cell-cell interface[37]. There is increasing evidence that GPCR signaling instructs the activation of actomyosin contractility in many morphogenetic processes[37–41] and reviewed in ref. 30 and ref. 42. In particular, *Drosophila* has five genes coding for GPCR serotonin receptors: 5HT1A, 5HT1B, 5HT2A, 5HT2B, and 5HT7[43,44]. Serotonin has been detected at its peak level at the onset of axis extension[28]. However, the function of serotonin and serotonin receptors during axis extension, which is driven by actomyosin contractility[32,33], remains elusive. Serotonin and GPCRs serotonin receptors have been identified in embryos of many organisms, including *Drosophila*[27] as described above, but whether they are involved in the regulation of actomyosin contractility has never been investigated.

In this study, we use invertebrate (*Drosophila*) and vertebrate (chicken) embryo model systems to investigate the potentially evolutionarily conserved role of serotonin and GPCR serotonin receptors during axis extension and regulation of cellular mechanics.

## Results

### 5HT2A is required for tissue flow, cell intercalation and MyoII activation in the *Drosophila* ectoderm

The *Drosophila* ectodermal tissue is on the lateral side of the embryo and undergoes convergence (along the Dorsal-Ventral; DV axis) and extension (along the Anterior-Posterior; AP axis) as the posterior pole of the embryo invaginates and moves towards the anterior[45,46] (Fig. 1a). We first asked whether 5HT2A is required for *Drosophila* axis extension using a null mutant based on homologous recombination (hereafter *5HT2A* −/−)[43,44]. In a previous study[29], germ-band extension defects were reported in a non-specific deficiency line which covers another gene.

Here, we re-examined this issue using a specific null mutant (*5HT2A*−/−) and analyzed the phenotypes at the tissue and cellular level. To determine whether 5HT2A mutation affects axis extension at the tissue level, we followed the progression of the posterior midgut in differential interference contrast (DIC) videos (Supplementary Fig. 1a, b and Supplementary Movie 1). We observed a 10–12 min delay in axis extension in maternal and zygotic 5HT2A mutant embryos compared to wild-type embryos (Supplementary Fig. 1a, b and Supplementary Movie 1). Consistently, local tissue extension, as measured by tracking the centroid distance between two cells was significantly reduced (Supplementary Fig. 2a, b).

Tissue extension is driven by cell intercalation which happens through T1 transition[32] and rosettes formation[47]. We tracked T1 events and found marked reduction (Fig. 1b, c) whereas rosettes formation was not significantly affected (Supplementary Fig. 2l). Cell intercalation is coordinated by spatiotemporal pattern of actomyosin contractility[48,49]. The planar polarized activity of MyoII at the junctions (MyoII is enriched in the vertical junctions)[32] and the planar polarized flow of the medial-apical MyoII[33] power polarized junctional remodelling[45] during cell intercalation. We next investigated the temporal evolution of MyoII distribution every 5 min, starting 30 min after

the cellularization front passes the nucleus or 20 min before the cell divisions in the dorsal ectoderm[45], using a tagged version of *Drosophila* MyoII-RLC (Sqh::mCherry). This time corresponds to the transition of the tissue movement towards the posterior. We observed an overall reduction of the junctional (Fig. 1d, Supplementary Fig. 2c and Supplementary Movie 2) and medial MyoII levels (Fig. 1d, g and Supplementary Movie 2) at all the time points. We examined the MyoII levels in DV-oriented and AP-oriented junctions, which allowed us to quantify the amplitude of polarity. MyoII levels at both categories of junctions were significantly reduced at all time points (Fig. 1e), however the amplitude of polarity, was reduced only initially (Fig. 1f). Medial-apical MyoII promotes E-cadherin clustering[50]. We observed reduced levels and polarity of E-cadherin (Supplementary Fig. 2e–g and Supplementary Movie 2), consistent with the reduction in medial MyoII levels. Taken together, 5HT2A recruits both junctional and medial MyoII and alters E-cadherin levels.

To confirm the phenotype of 5HT2A mutation, we next studied gain of 5HT2A function by overexpressing the wild-type 5HT2A receptors. Assuming that G-proteins and downstream effectors are abundant, overexpression of 5HT2A would be expected to increase both junctional and medial MyoII levels. Consistently, we observed an increase in the junctional MyoII levels (Fig. 1h, Supplementary Fig. 2d and Supplementary Movie 3). However, the medial MyoII levels did not increase (Fig. 1h, k). On closer inspection, we found progressive enrichment of MyoII specifically at the DV (vertical) junctions (Fig. 1h, i). The amplitude of MyoII polarity in the wild-type embryos tends to decrease over time[48,49]. Interestingly, it significantly increased after 5HT2A overexpression compared to the wild-type embryos (Fig. 1j). In line with this, we observed more aligned DV-oriented junctions forming supracellular cables that are indistinguishable from the parasegment boundaries in 5HT2A overexpressing embryos (Fig. 1h and Supplementary Fig. 2h), indicating that the majority of DV-oriented junctions did not shrink; consistent with this hypothesis, there was a significant reduction in cumulative T1 events (Fig. 1l, m). However, local tissue extension was not significantly affected (Supplementary Fig. 2i) and cumulative rosettes were higher (Supplementary Fig. 2j, k). While, in control, T1s appear first, followed by rosettes (after 10 min) (Fig. 1m and Supplementary Fig. 2k)[36,47]. In 5HT2A gain-of-function, rosettes appear 5 min earlier. This suggests that the reduced T1 events in 5HT2A overexpression are compensated for by increased rosette formation, resulting in no significant effect on tissue extension.

Taken together, these results suggest a dose-dependent effect of 5HT2A on MyoII levels, polarity and spatial patterning. At zero levels of 5HT2A (in 5HT2A mutants), junctional and medial MyoII levels were reduced and polarization was largely unaffected, whereas overexpression of 5HT2A led to enrichment of MyoII specifically at the DV-oriented junctions, giving rise to hyper-polarization. Among the GPCRs known to activate MyoII in the ectoderm namely Smog[38] and Cirl[37], 5HT2A is the only GPCR that hyper-polarizes junctional MyoII when overexpressed. This raises the question of how the overexpression of 5HT2A hyper-polarizes the junctional MyoII. One possibility is that the 5HT2A localization is polarized and hence this imparts polarization to MyoII. To investigate the localization of 5HT2A, we generated a C-terminally tagged 5HT2A::mNeonGreen construct (Supplementary Fig. 3a) under the *sqh* promoter to have homogenous overexpression. MyoII was specifically recruited to the DV-oriented junctions in 5HT2A::mNeonGreen overexpressing embryos, resulting in hyper-polarization similar to wild-type receptor overexpression (Supplementary Fig. 3b–d). 5HT2A::mNeonGreen was localized throughout the cell membrane as well as in the cytoplasmic organelle (Supplementary Fig. 3e, f). These cytoplasmic organelles were endocytic vesicles, as they co-localized with dextran filled mobile endocytic vesicles (Supplementary Fig. 8a, a'). Significantly, the membrane localization of 5HT2A::mNeonGreen was not polarized (Supplementary Fig. 3e, f).

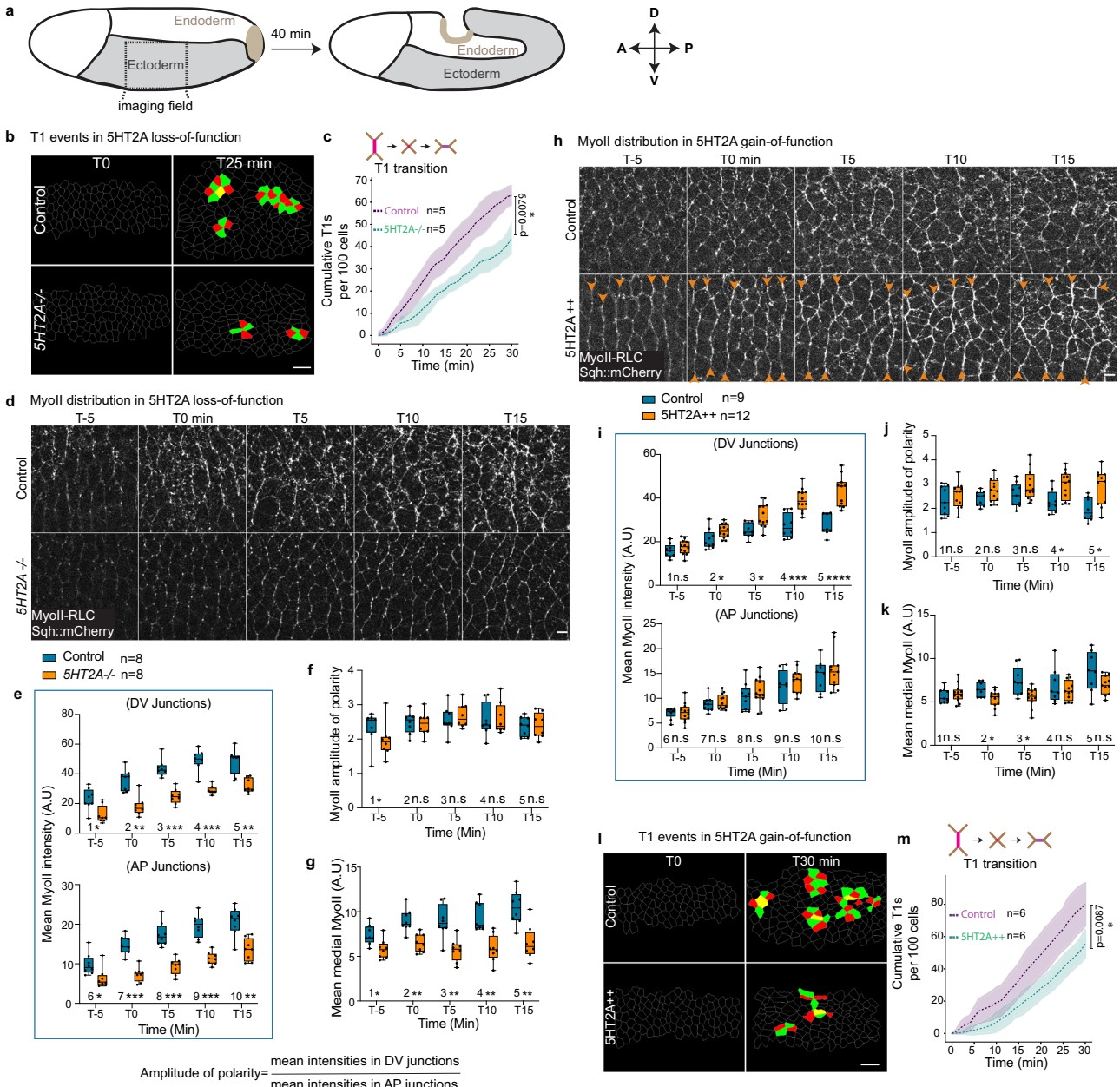

**Fig. 1 | 5HT2A is required for cell intercalation and MyoII accumulation in the *Drosophila* ectoderm. a** Ectoderm morphogenesis during *Drosophila* axis extension. **b**, **c** T1 events in 5HT2A loss-of-function. Representative images of T1s in control (top panels) and 5HT2A mutant (*SHT2A*−/−) (bottom panels) taken at T0 (left) and T25 min (right) (**b**) and quantification of cumulative T1s in the respective conditions (**c**). The center dashed line is mean and the error bands are the standard deviation. Statistical significance was calculated by two-tailed Mann−Whitney *U* test at 30 min (*P ≤ 0.05). **d**−**g** MyoII distribution in 5HT2A loss-of-function in the region outlined in **a**. Live images of MyoII every 5 min interval, upper panels: control, lower panels: *SHT2A*−/− (**d**). Quantification of mean MyoII intensities in DV-oriented and AP-oriented junctions (**e**), amplitude of polarity (**f**) and mean medial-apical MyoII intensity (**g**) over time. **h**−**k** MyoII distribution in 5HT2A overexpression (5HT2A ++). Live images of MyoII over-time, upper panels: control, lower panels: 5HT2A ++, orange arrowheads showing the hyper-polarization of junctional MyoII (**h**). Quantification of MyoII intensities in DV-oriented and AP oriented junctions (**i**), amplitude of polarity (**j**) and mean medial MyoII intensities (**k**). **l**, **m** T1 events in 5HT2A gain-of-function. Representative images of the T1 events taken at T0 (left) and T30 min (right) for control (upper panels) and 5HT2A ++ (lower panels) (**l**) and

quantification of cumulative T1 events in the respective conditions (**m**). The center dashed line is the mean and the error bands are the standard deviation. Statistical significance was calculated by two-tailed Mann−Whitney *U* test at 30 min (*P ≤ 0.05). In the box plots in (**e**, **f**, **g**, **i**, **j**, **k**), the line in the middle is plotted at the median. The box extends from the 25th to 75th percentiles. The whiskers indicate minimum and maximum values. Statistics: ns $P > 0.05$, *$P ≤ 0.05$, **$P ≤ 0.005$, ***$P ≤ 0.0005$, ****$P ≤ 0.00005$ from two-tailed Mann−Whitney test. *P* values in (**e**): 1 ($P = 0.015$), 2 ($P = 0.001$), 3 ($P = 1.55 \times 10^{-4}$), 4 ($P = 3.11 \times 10^{-4}$), 5 ($P = 0.0023$), 6 ($P = 0.015$), 7 ($P = 1.55 \times 10^{-4}$), 8 ($P = 1.55 \times 10^{-4}$), 9 ($P = 1.55 \times 10^{-4}$), 10 ($P = 0.003$). *P* values in (**f**): 1 ($P = 0.021$), 2 ($P = 0.798$), 3 ($P = 0.645$), 4 ($P = 0.959$), 5 ($P = 0.959$). *P* values in (**g**): 1 ($P = 0.015$), 2 ($P = 0.001$), 3 ($P = 0.002$), 4 ($P = 6.22 \times 10^{-4}$), 5 ($P = 0.003$). *P* values in (**i**): 1 ($P = 0.219$), 2 ($P = 0.034$), 3 ($P = 0.007$), 4 ($P = 4.76 \times 10^{-4}$), 5 ($P = 2.6 \times 10^{-5}$), 6 ($P > 0.999$), 7 ($P = 0.464$), 8 ($P = 0.508$), 9 ($P = 0.427$), 10 ($P = 0.657$). *P* values in (**j**): 1 ($P = 0.554$), 2 ($P = 0.082$), 3 ($P = 0.148$), 4 ($P = 0.016$), 5 ($P = 0.016$). *P* values in (**k**): 1 ($P = 0.651$), 2 ($P = 0.041$), 3 ($P = 0.012$), 4 ($P = 0.851$), 5 ($P = 0.109$). *n* = number of embryos. Scale bars in (**b**, **l**) 10 μm, in (**d**, **h**) 5 μm. Source data are provided as a Source Data file.

## Serotonin signaling is permissive in the ectoderm

Since the localization of the receptor is not polarized, we next investigated the possibility that the receptor activation by the ligand is polarized. Tryptophan hydroxylase (hereafter Trh), is the rate-limiting enzyme involved in the biogenesis of serotonin (Fig. 2a), the bonafide ligand recognized by serotonin receptors and present at the beginning of germ band extension[28]. We therefore investigated the null mutant $Trh^{01}$ [44], in which the enzyme activity is impaired. Maternal and zygotic $Trh^{01}$ mutant embryos develop normally, and this mutation has a mild effect on germ-band extension (Supplementary Fig. 1c, d). Maternal and zygotic knockout of Trh resulted in the specific reduction of the medial-apical MyoII levels (Fig. 2b, c and Supplementary Movie 4) with no significant changes in the junctional MyoII levels (Fig. 2b, d and Supplementary Movie 4). The reduction of medial MyoII was similar to that in 5HT2A mutant embryos (Fig. 1d, g). This suggests that serotonin is not required for junctional signaling under physiological conditions and that 5HT2A signaling at the junctions is serotonin independent, or that another ligand compensates for absence of serotonin. Medial apical MyoII levels are sensitive to the ligand dosage (overexpression of another ligand Fog which partially signals through Smog in the ectoderm, results in the hyperactivation of apical MyoII in the ectoderm[38,51]. To investigate the dose response of the serotonin, we studied the gain of Trh function by overexpressing wild-type Trh enzyme. We presume that this leads to elevated serotonin levels as reported elsewhere[52,53]. Interestingly, MyoII levels in the DV-oriented junctions were higher at all time points and the amplitude of polarity was significantly increased (Fig. 2e–g and Supplementary Movie 5), whereas medial MyoII levels did not increase at most time points (Fig. 2e, h). These results reveal the dose-dependent effect of serotonin in the spatial control of MyoII recruitment.

Overexpression of both the ligand and the receptor is similar in hyper-polarizing junctional MyoII. Loss of both the ligand and the receptor is similar in reduction of medial MyoII levels. To further investigate whether serotonin functions via 5HT2A, we first asked if serotonin requires 5HT2A to hyper-activate junctional MyoII when Trh is overexpressed. To test this, we overexpressed Trh in 5HT2A mutant embryos and observed a reduction in both junctional and medial MyoII levels, similar to that observed in 5HT2A mutant embryos, and a loss of hyper-polarization (Fig. 2i–k and Supplementary Movie 7). We then asked whether 5HT2A mediated hyper-polarization requires serotonin. Following the overexpression of 5HT2A in $Trh^{01}$ mutant embryos, junctional MyoII hyper-polarization was no longer observed (Fig. 2l–n and Supplementary Movie 6). Junctional MyoII levels were similar to WT embryos and medial MyoII levels were significantly reduced (Fig. 2l, m, o). These data are similar to the $Trh^{01}$ mutant embryos (Fig. 2b–d). Taken together, these data argue that serotonin exerts its function through 5HT2A.

In conclusion, at physiological levels, serotonin and 5HT2A are essential for medial apical MyoII recruitment. At the junctions, serotonin is not essential at physiological levels to control MyoII (Fig. 2b, d), presumably due to redundancy with another ligand. However, serotonin functions via 5HT2A when either the ligand (serotonin) or the receptor is overexpressed (Fig. 2i–o) and such redundancy is no longer apparent. The fact that reduced serotonin synthesis does not perturb polarity at junctions also indicates that serotonin signaling is permissive in the ectoderm.

## 5HT2A sets the magnitude of Toll/Cirl polarity signaling

What could be the additional ligand recognized by 5HT2A at the junctions? Three Toll receptors, Toll 2, 6 and 8, impart the polarization of junctional MyoII[36,37]. Toll receptor, for example Toll-8 and the adhesion GPCR Cirl/Latrophilin interact to form a complex, mutually attract each other and mediate MyoII polarization at the cell-cell interface (Fig. 3a and ref. 37).

We wanted to know whether Toll receptors instruct 5HT2A mediated hyper-polarization. To this end, we knocked-down Toll 2,6,8 (by injecting *toll-2,6,8* RNAi) in 5HT2A overexpressing embryos and found a similar MyoII levels and polarity as in *toll-2,6,8* RNAi injected WT embryos (Fig. 3b–d and Supplementary Movie 8). We no longer observed hyper-polarization of MyoII (Fig. 3b, d and Supplementary Movie 8), suggesting that the Toll receptors are required for 5HT2A mediated amplification of MyoII polarity. Concomitantly, 5HT2A::m-NeonGreen membrane localization was depleted following the knockdown of Toll-2,6,8 and accumulated in intracellular vesicles (Supplementary Fig. 4a, b), junctional MyoII levels and polarity were consistently reduced (Supplementary Fig. 4a, c). This suggests that, Toll receptors are required to recruit or stabilize 5HT2A to the membrane.

Toll-8 interacts with Cirl and recruits MyoII at the cell-cell interface[37]. We therefore investigated whether 5HT2A transduces Cirl polarized signaling. Both Cirl and 5HT2A null mutants were homozygous viable, however double-mutants were homozygous embryonic lethal. Interestingly, we observed that the heterozygous 5HT2A mutation enhanced the phenotype of the homozygous Cirl mutation with respect to the wild-type embryos (Fig. 3e–g). The levels of junctional MyoII (Fig. 3e, f) and the local tissue extension (Fig. 3g) were reduced compared to the wild-type embryos in both the homozygous Cirl mutant (*Cirl−/−*) and homozygous 5HT2A mutant (*5HT2A−/−*) embryos (Fig. 3e, f and tissue extension in 5HT2A mutant embryos in Supplementary Fig. 2a, b) alone; while in the homozygous Cirl mutant and heterozygous 5HT2A mutant embryos (*Cirl −/− 5HT2A−/+*), MyoII levels and local tissue extension were further reduced (Fig. 3e–g and Supplementary Movie 9). In contrast, heterozygous Cirl mutation (*Cirl −/+*) did not enhance homozygous 5HT2A mutation junctional MyoII phenotype (Fig. 3e, f). Taken together, these data suggest a synergistic or additive effect of Cirl and 5HT2A on junctional MyoII recruitment and tissue extension and that 5HT2A levels are critical.

To investigate whether 5HT2A signals independently of Cirl, we overexpressed 5HT2A in Cirl mutant embryos (Fig. 3h–k). Strikingly, ectopic 5HT2A rescued the low levels of MyoII in Cirl mutant embryos, but not the planar polarity of MyoII (Fig. 3h–k and Supplementary Movie 10). In Cirl mutants, junctional MyoII levels were reduced at all time points (Fig. 3h, i), while polarity was significantly reduced only initially and remained low at later time points (Fig. 3j), and medial MyoII levels were significantly reduced at all time points (Fig. 3h, k). When 5HT2A was overexpressed in Cirl mutant embryos, junctional MyoII levels were similar to wild-type embryos (Fig. 3h, i) and medial MyoII levels were partially rescued (Fig. 3h, k). Surprisingly, the amplitude of polarity remained low at most time points as in Cirl mutant embryos (Fig. 3j). This indicates that 5HT2A quantitatively controls the MyoII levels and extent of the Cirl-dependent planar polarity cue. While Cirl/Toll impart polarization per se, 5HT2A controls the magnitude of this effect but not polarity itself. Overall, we conclude that the 5HT2A forms a distinct signaling module that regulates the MyoII levels, while Toll/Cirl serves as polarizing signaling module.

## 5HT2A activates junctional Rho1/MyoII through Gβ13f/Gγ1 and Dp114RhoGEF

GPCR signaling activates MyoII through the Rho1 pathway (Fig. 4a and refs. 38,39). We wanted to know whether 5HT2A signals through this pathway. We monitored the Rho1 activity over time using the Rho1 biosensor[54] in 5HT2A mutant embryos, and found the levels of Rho1-GTP were significantly reduced at both the junctions and medial-apically at all the time points (Fig. 4b–d and Supplementary Movie 11). The Rho1 amplitude of polarity was not significantly reduced (Fig. 4e). In 5HT2A gain-of-function embryos, junctional Rho1-GTP levels were initially not much different than in wild-type embryos, while at later time points, Rho1-GTP signals tended to decrease in the wild-type, whereas in 5HT2A overexpression, it

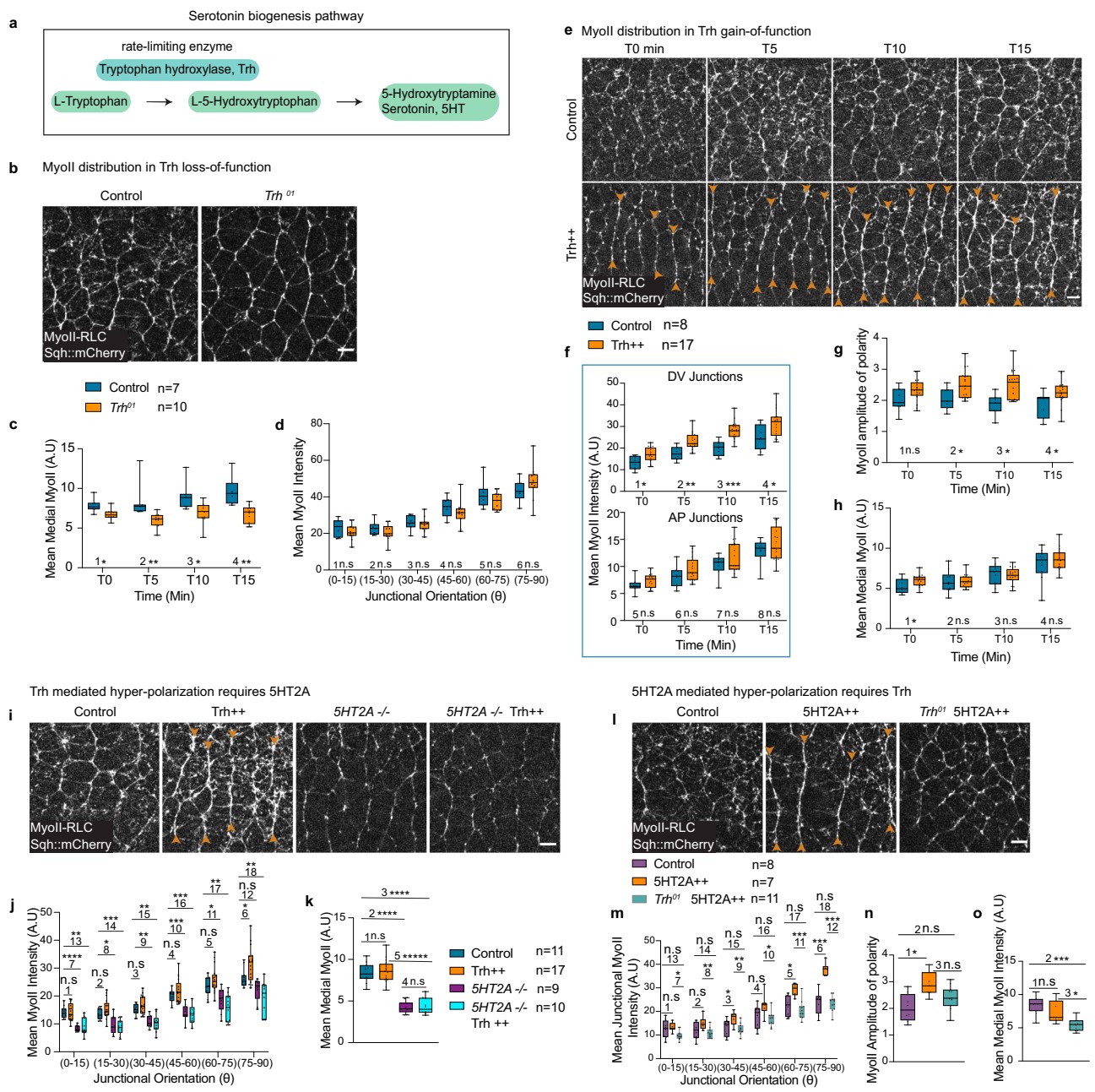

**Fig. 2 | Serotonin signaling is permissive in the ectoderm. a** Schematic of serotonin biogenesis, tryptophan hydroxylase (Trh) is the rate-limiting enzyme. **b**–**d**, **b** Snapshot of MyoII in Trh loss-of-function (*Trh^{01}* mutant); control (left panel) and *Trh^{01}* (right panel). Quantification of mean medial-apical MyoII levels over-time (**c**) and an example of the distribution of junctional MyoII intensities in different junctional orientations for different conditions (**d**). **e**–**h** Time-lapse of MyoII in Trh gain-of-function (Trh + +), top panels: control, lower panels: Trh + +, orange arrowheads indicate hyper-polarization of junctional MyoII (**e**). Quantification of MyoII intensities over-time in DV and AP oriented junctions (**f**), amplitude of polarity (**g**), and medial MyoII intensities (**h**). **i**–**k** Trh + + and *5HT2A^{−/−}*. **i** MyoII images in different conditions (from left to right respectively): control, Trh + + (orange arrowheads indicate hyper-polarization), *5HT2A-/-* and *5HT2A-/-* Trh + +. Quantification of junctional (**j**), and medial MyoII (**k**) intensities in different conditions. **l**–**o** *Trh^{01}* and 5HT2A + +. MyoII in control (left panel), 5HT2A + + (middle panel, orange arrowheads indicate hyper-polarization) and *Trh^{01}* 5HT2A + + (right panel) (**l**), quantification of junctional MyoII intensities (**m**), amplitude of polarity (**n**), and medial MyoII (**o**) in different genotypes. In the box plots in (**c**, **d**, **f**, **g**, **h**, **j**, **k**, **m**, **n**, **o**), the line in the middle is plotted at the median. The box extends from the 25th to 75th percentiles. The whiskers indicate minimum and

maximum values. Statistics: ns: $P > 0.05$, *$P \leq 0.05$, **$P \leq 0.005$, ***$P \leq 0.0005$, ****$P \leq 0.00005$, *****$P \leq 0.000005$ from two-tailed Mann–Whitney test. *P* values in (**c**): 1 ($P = 0.01$), 2 ($P = 0.0004$), 3 ($P = 0.014$), 4 ($P = 0.0007$). *P* values in (**d**): 1 ($P = 0.375$), 2 ($P = 0.285$), 3 ($P = 0.596$), 4 ($P = 0.211$), 5 ($P = 0.328$), 6 ($P = 0.285$). *P* values in (**f**): 1 ($P = 0.013$), 2 ($P = 0.002$), 3 ($P = 0.000055$), 4 ($P = 0.022$), 5 ($P = 0.157$), 6 ($P = 0.238$), 7 ($P = 0.475$), 8 ($P = 0.475$). *P* values in (**g**): 1 ($P = 0.114$), 2 ($P = 0.024$), 3 ($P = 0.005$), 4 ($P = 0.03$). *P* values in (**h**): 1 ($P = 0.035$), 2 ($P = 0.711$), 3 ($P = 0.537$), 4 ($P = 0.786$). *P* values in (**j**): 1 ($P = 0.963$), 2 ($P = 0.134$), 3 ($P = 0.264$), 4 ($P = 0.43$), 5 ($P = 0.147$), 6 ($P = 0.029$), 7 ($P = 1.2 \times 10^{-5}$), 8 ($P = 0.01$), 9 ($P = 0.001$), 10 ($P = 2.26 \times 10^{-4}$), 11 ($P = 0.038$), 12 ($P = 0.056$), 13 ($P = 0.001$), 14 ($P = 3.8 \times 10^{-4}$), 15 ($P = 0.001$), 16 ($P = 1.7 \times 10^{-4}$), 17 ($P = 0.001$), 18 ($P = 0.005$). *P* values in (**k**): 1 ($P = 0.853$), 2 ($P = 1.2 \times 10^{-5}$), 3 ($P = 6 \times 10^{-6}$), 4 ($P = 0.842$), 5 ($P < 1 \times 10^{-6}$). *P* values in (**m**): 1 ($P = 0.694$), 2 ($P = 0.232$), 3 ($P = 0.04$), 4 ($P = 0.121$), 5 ($P = 0.006$), 6 ($P = 3.11 \times 10^{-4}$), 7 ($P = 0.008$), 8 ($P = 0.004$), 9 ($P = 0.004$), 10 ($P = 0.011$), 11 ($P = 4.4 \times 10^{-4}$), 12 ($P = 6.3 \times 10^{-5}$), 13 ($P = 0.237$), 14 ($P = 0.442$), 15 ($P = 0.6$), 16 ($P = 0.657$), 17 ($P = 0.6$), 18 ($P = 0.6$). *P* values in (**n**): 1 ($P = 0.009$), 2 ($P = 0.091$), 3 ($P = 0.104$). *P* values in (**o**): 1 ($P = 0.281$), 2 ($P = 0.0005$), 3 ($P = 0.006$). $n$ = number of embryos. Scale bars 5 µm. Source data are provided as a Source Data file.

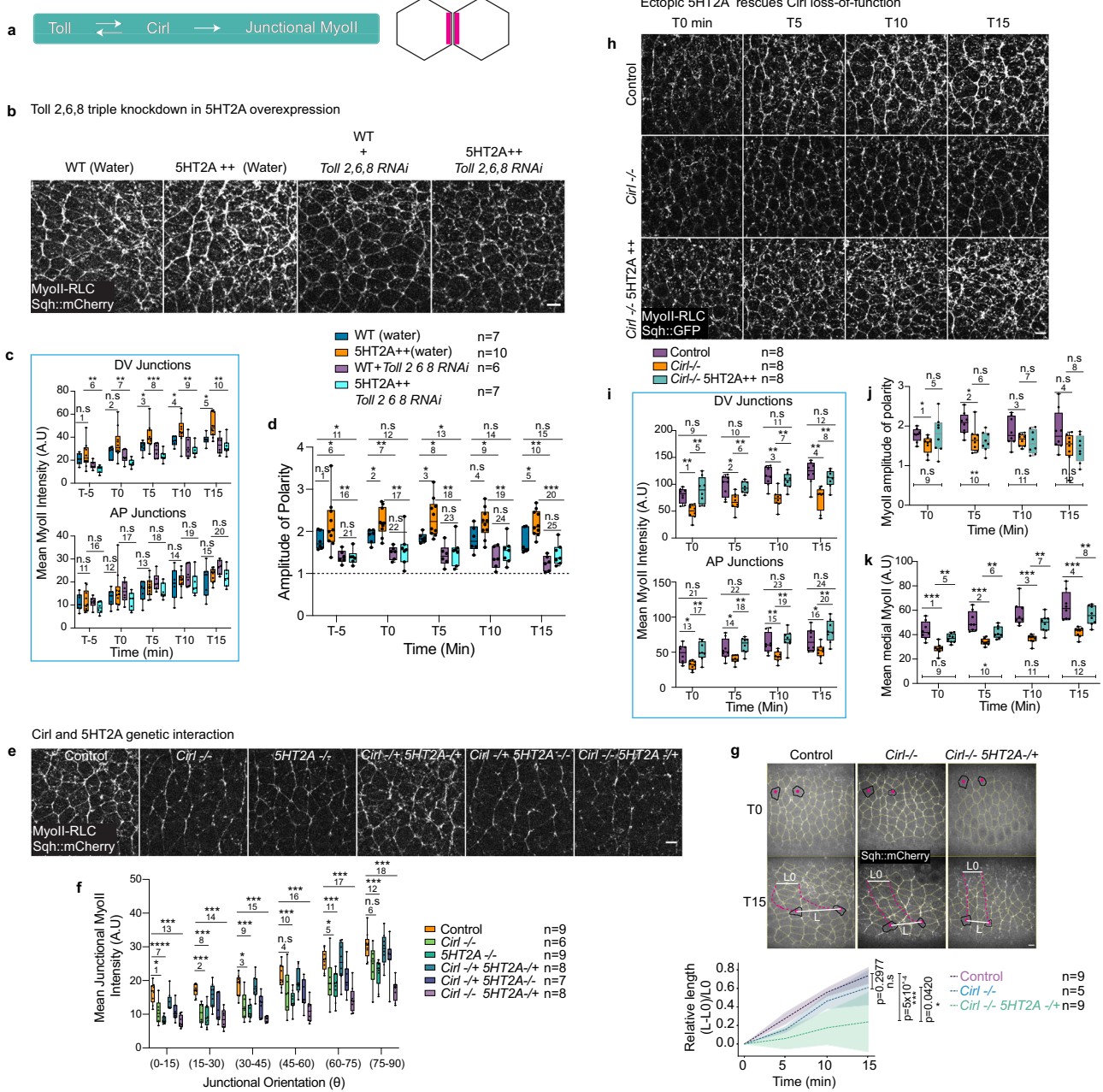

remained significantly higher in all junctional orientations (Fig. 4f, g and Supplementary Movie 12), in contrast, MyoII was enriched specifically at DV-oriented junctions (Fig. 1h, i). We observed a slight but not significant increase in the Rho1-GTP polarity (Fig. 4h). To investigate this difference, we hypothesized that Rho1 signaling is further amplified downstream and studied the downstream effectors of Rho1: Rho-associated kinase (Rok) and myosin phosphatase. We found that the overexpression of 5HT2A represses myosin phosphatase (Supplementary Fig. 5a, b and Supplementary Movie 13), which inhibits the de-phosphorylation of MyoII as revealed by monitoring the junctional recruitment of the myosin-binding subunit of myosin phosphatase (MBS::GFP). This suggests that overexpression of 5HT2A enhances the stabilization of junctional MyoII, leading to hyper-polarization. To confirm that the emergence of hyper-polarization of junctional MyoII is dependent on Rok activity, we injected the Rok inhibitor H-1152 compound and we observed that upon this treatment the hyper-polarization of MyoII was lost (Supplementary Fig. 5c), confirming that the 5HT2A

mediated hyperactivation of junctional MyoII requires Rok activity. Taken together, 5HT2A controls MyoII through the Rho1 signaling pathway.

Dp114RhoGEF compartmentalizes Rho1 activity at cell junctions (Fig. 4a and refs. 39,55). We asked whether the 5HT2A mediated activation of junctional Rho1/MyoII is dependent on Dp114RhoGEF. The endogenous levels of Dp114RhoGEF were significantly reduced in 5HT2A mutant embryos (Fig. 4i, j). Conversely, we observed elevated levels of Dp114RhoGEF at the junctions in 5HT2A overexpressing embryos (Fig. 4k, l). Furthermore, overexpression of the 5HT2A in Dp114RhoGEF knockdown embryos did not result in hyper-polarization of junctional MyoII (Fig. 4m–o). Junctional MyoII levels and the amplitude of polarity were reduced in a manner similar to Dp114RhoGEF knockdown embryos (Fig. 4m–o). These data confirm that the 5HT2A activates junctional MyoII through Dp114RhoGEF.

The heterotrimeric G-proteins Gα$_{12/13}$ Gβ13f/Gγ1 transduce GPCR signaling to modulate apical and junctional Rho1/MyoII activation (Fig. 4a and refs. 38,39). Gα-GTP activates apical Rho1/MyoII while

**Fig. 3 | 5HT2A amplifies Toll/Cirl polarity cue. a** Schematic showing Toll/Cirl mutual attraction recruiting MyoII at the cell-cell interface. **b**–**d** 5HT2A over-expression and *Toll-2,6,8* knockdown. **b** MyoII images in different conditions (from left to right respectively): water injected WT, water injected 5HT2A + +, *Toll 2,6,8 RNAi* injected WT, and *Toll 2,6,8 RNAi* injected 5HT2A + + embryos. Quantification of MyoII intensities in DV and AP oriented junctions over-time (**c**), amplitude of polarity (**d**) in different conditions. **e**, **f** Genetic interaction between Cirl and 5HT2A. **e** MyoII snapshots in different conditions (from left to right respectively): control, *Cirl−/−*, *5HT2A−/−*, *Cirl−/+ 5HT2A−/+*, *Cirl-/+ 5HT2A−/−* and *Cirl−/− 5HT2A−/+*. **f** Quantification of junctional MyoII intensities in different genotypes. **g** Quantification of local tissue extension. Representative images are average pro-jection of Sqh::mCherry. The centroid of the cells outlined in dark was tracked every 5 min for 15 min starting 30 min after the cellularization front passed the nucleus, pink dotted lines are the track of the centroids. The relative length (L-L0)/L0 is plotted at the bottom. Center dashed line is mean and error bands are stan-dard deviation. Statistical significance was calculated by two-tailed Mann–Whitney *U* test at 15 min (ns *P* > 0.05, *\*P* ≤ 0.05, \*\*\**P* ≤ 0.0005). **h**–**k** 5HT2A overexpression in *Cirl-/-* showing that ectopic 5HT2A rescues MyoII levels in *Cirl−/−*. **h** MyoII images at different time points (left to right) in different genotypes (top to bottom): control (top panel), *Cirl−/−* (middle panel) and *Cirl-/- 5HT2A + +* (bottom panel). Quantifi-cation of time traces of MyoII intensities in DV and AP oriented junctions (**i**), amplitude of polarity (**j**), and medial MyoII intensities (**k**) in different genotypes. In the box plots in (**c**, **d**, **f**, **i**, **j**, **k**), the line in the middle is plotted at the median. The box extends from the 25th to 75th percentiles. The whiskers indicate minimum and maximum values. Statistics: ns *P* > 0.05, *\*P* ≤ 0.05, \*\**P* ≤ 0.005, \*\*\**P* ≤ 0.0005,

\*\*\*\**P* ≤ 0.00005 from two-tailed Mann−Whitney test. *P* values in (**c**): 1 (*P* = 0.635), 2 (*P* = 0.368), 3 (*P* = 0.031), 4 (*P* = 0.022), 5 (*P* = 0.011), 6 (*P* = 0.005), 7 (*P* = 0.0007), 8 (*P* = 0.0004), 9 (*P* = 0.005), 10 (*P* = 0.002), 11 (*P* = 0.93), 12 (*P* = 0.479), 13 (*P* = 0.669), 14 (*P* = 0.536), 15 (*P* = 0.475), 16 (*P* = 0.256), 17 (*P* = 0.126), 18 (*P* = 0.364), 19 (*P* = 0.417), 20 (*P* = 0.536). *P* values in (**d**): 1 (*P* = 0.098), 2 (*P* = 0.015), 3 (*P* = 0.007), 4 (*P* = 0.073), 5 (*P* = 0.031), 6 (*P* = 0.009), 7 (*P* = 0.004), 8 (*P* = 0.026), 9 (*P* = 0.026), 10 (*P* = 0.002), 11 (*P* = 0.015), 12 (*P* = 0.051), 13 (*P* = 0.035), 14 (*P* = 0.101), 15 (*P* = 0.073), 16 (*P* = 0.001), 17 (*P* = 0.004), 18 (*P* = 0.002), 19 (*P* = 0.002), 20 (*P* = 0.0002), 21 (*P* = 0.818), 22 (*P* = 0.628), 23 (*P* = 0.534), 24 (*P* = 0.445), 25 (*P* = 0.181). *P* values in (**f**): 1 (*P* = 0.012), 2 (*P* = 0.0004), 3 (*P* = 0.008), 4 (*P* = 0.088), 5 (*P* = 0.036), 6 (*P* = 0.088), 7 ($P = 4.1 \times 10^{-5}$), 8 ($P = 1.65 \times 10^{-4}$), 9 ($P = 8.2 \times 10^{-5}$), 10 ($P = 2.88 \times 10^{-4}$), 11 ($P = 4.94 \times 10^{-4}$), 12 ($P = 4.94 \times 10^{-4}$), 13 ($P = 8.2 \times 10^{-5}$), 14 ($P = 8.2 \times 10^{-5}$), 15 ($P = 8.2 \times 10^{-5}$), 16 ($P = 1.65 \times 10^{-4}$), 17 ($P = 3.29 \times 10^{-4}$), 18 ($P = 8.2 \times 10^{-5}$). *P* values in (**i**): 1 (*P* = 0.002), 2 (*P* = 0.01), 3 ($P = 6.22 \times 10^{-4}$), 4 (*P* = 0.005), 5 (*P* = 0.002), 6 (*P* = 0.003), 7 (*P* = 0.001), 8 (*P* = 0.005), 9 (*P* = 0.442), 10 (*P* = 0.505), 11 (*P* = 0.235), 12 (*P* = 0.2), 13 (*P* = 0.021), 14 (*P* = 0.021), 15 (*P* = 0.003), 16 (*P* = 0.05), 17 (*P* = 0.005), 18 (*P* = 0.002), 19 (*P* = 0.005), 20 (*P* = 0.002), 21 (*P* = 0.574), 22 (*P* = 0.442), 23 (*P* = 0.234), 24 (*P* = 0.195). *P* values in (**j**): 1 (*P* = 0.021), 2 (*P* = 0.021), 3 (*P* = 0.328), 4 (*P* = 0.13), 5 (*P* = 0.328), 6 (*P* = >0.999), 7 (*P* = 0.798), 8 (*P* = 0.878), 9 (*P* = 0.959), 10 (*P* = 0.005), 11 (*P* = 0.234), 12 (*P* = 0.105). *P* values in (**k**): 1 ($P = 3.11 \times 10^{-4}$), 2 ($P = 1.55 \times 10^{-4}$), 3 ($P = 1.55 \times 10^{-4}$), 4 ($P = 1.55 \times 10^{-4}$), 5 (*P* = 0.002), 6 (*P* = 0.001), 7 (*P* = 0.002), 8 (*P* = 0.001), 9 (*P* = 0.083), 10 (*P* = 0.028), 11 (*P* = 0.105), 12 (*P* = 0.234). *n* = number of embryos. Scale bars 5 μm. Source data are provided as a Source Data file.

---

Gβ13f/Gγ1 heterodimer activate junctional Rho1/MyoII. Gβ13f/Gγ1 overexpression hyper-activates junctional Rho1/MyoII[39] similar to 5HT2A overexpression. If 5HT2A signals through Gβ13f/Gγ1, removal of 5HT2A and overexpression of Gβ13f/Gγ1 should have no effect unless 5HT2A is required for the ectopic Gβ13f/Gγ1 activation. We therefore asked whether ectopic Gβ13f/Gγ1 mediated hyper-activation of MyoII requires 5HT2A. To examine this, we over-expressed Gβ13f/Gγ1 in the 5HT2A mutant embryos. Hyperactivation was lost and MyoII levels were reduced to inter-mediate levels between WT and 5HT2A mutant embryos (Fig. 4p, q). These data suggest that 5HT2A is required for Gβ13f/Gγ1 dependent activation of junctional MyoII. Given that Dp114RhoGEF mediates Gβ13f/Gγ1 signaling to activate junctional Rho1/MyoII (Fig. 4a and ref. 39) and that 5HT2A requires Dp114RhoGEF (Fig. 4m–o), we conclude that 5HT2A signals through Gβ13f/Gγ1 and Dp114RhoGEF to activate junctional Rho1/MyoII. The mammalian ortholog of 5HT2A is known to activate PLC through the canonical GαQ sig-naling pathway that leads to accumulation of IP3, DAG and PKC[56,57]. Here we find that 5HT2A activates Rho1/MyoII through a non-canonical Gβ13f/Gγ1 and Dp114RhoGEF signaling pathway.

### 5HT2B inhibits MyoII in the ectoderm and requires 5HT2A

All the experiments point to a key quantitative role of 5HT2A signaling in tuning the junctional activation of MyoII, whose polarization is imparted specifically by Toll/Cirl. This begs the question of how the level of signaling by 5HT2A is regulated. It has been reported that the mammalian 5HT2 class receptors (5HT2A, 5HT2B and 5HT2C) interact to form heterodimers[58], that bias the signaling of one protomer over the other. Since *Drosophila* lacks 5HT2C, we examined germ band extension in 5HT2B null mutant embryos[43,44] and observed a delay of almost 15 min (Supplementary Fig. 1a, b and Supplementary Movie 1). Strikingly, MyoII levels were higher at the junctions as well as in the medial pool (Supplementary Fig. 6a, b, d and Supplementary Movie 14). Consistently, we observed a similar phenotype in embryos expressing short-hairpin RNA (shRNA) against 5HT2B (Supplementary Fig. 6e, f, h) and following injection of 5HT2B double-stranded RNA (5HT2B dsRNA) (Fig. 5a–c and Supplementary Movie 15). These data suggest a repression of MyoII enrichment by this receptor. Amplitude of MyoII polarity was not affected (Fig. 5d and Supplementary Fig. 6c, g).

To investigate how 5HT2B inhibits MyoII accumulation, we con-sidered the possibility that the inhibitory effect emerged from an interaction between 5HT2A and 5HT2B, consistent with reports in other systems[58]. One could envision two possible interactions (Fig. 5e): 1) 5HT2A inhibits 5HT2B and 5HT2B inhibits MyoII per se. 2) 5HT2B inhibits 5HT2A and 5HT2A activates MyoII per se. Removal of the individual receptors does not distinguish between these two hypoth-eses, however, removal of both the receptors is predicted to have different outcomes. According to the first hypothesis, in the absence of 5HT2A, 5HT2B becomes constitutively active and represses MyoII. In double mutants, this will be similar to removing the repressor 5HT2B alone, with higher levels of MyoII. According to the second hypothesis, when 5HT2B is removed, 5HT2A becomes hyper-activated and acti-vates more MyoII. But when both are removed, the situation will be similar to removing the activator 5HT2A with reduced levels of MyoII. To test these predictions, we injected 5HT2B dsRNA into the 5HT2A mutant embryos. MyoII levels were reduced as in the 5HT2A mutant embryos (Fig. 5f–h and Supplementary Movie 16), supporting the second hypothesis that 5HT2B inhibits 5HT2A and 5HT2A activates MyoII (Fig. 5i). This argues that 5HT2B requires 5HT2A to repress MyoII recruitment both at the junctions and medially.

### 5HT2A and 5HT2B interact to form heterocomplexes and undergo endocytosis

The above study demonstrated a genetic interaction between 5HT2A and 5HT2B. To further investigate whether 5HT2A and 5HT2B physi-cally interact to form heterocomplexes, we used scanning Fluores-cence Cross-Correlation Spectroscopy (sFCCS)[59,60] (Fig. 5j–o) to probe co-diffusion of 5HT2A and 5HT2B at the membrane in vivo (Fig. 5j). We developed a C-terminally tagged 5HT2B::mCherry construct (Supple-mentary Fig. 7a). The fluorescently labelled 5HT2B behaved like the wild-type receptor, as junctional MyoII remained unaffected and medial MyoII levels were reduced in both wild-type (Supplementary Fig. 7b–d) and fluorescently labelled (Supplementary Fig. 7e–g) receptors overexpression. 5HT2B::mCherry was localized to both the membrane and cytoplasmic organelles (Supplementary Fig. 7h, i). When co-expressed, 5HT2A::mNeonGreen and 5HT2B::mCherry co-localized both at the membrane and in cytoplasmic organelles (Sup-plementary Fig. 8c, c'). We performed sFCCS in embryos co-expressing 5HT2A::mNeonGreen and 5HT2B::mCherry and in negative control

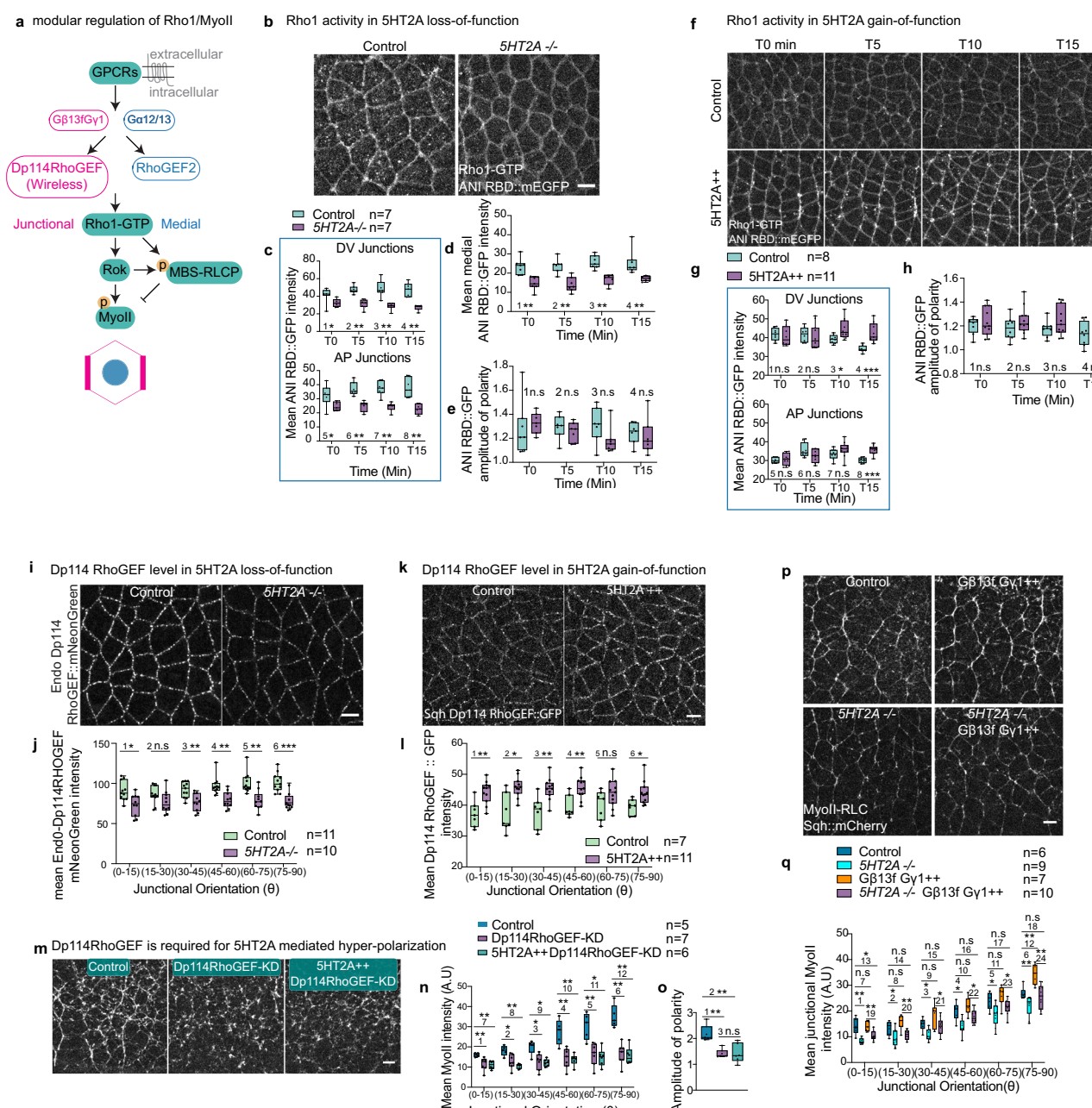

**Fig. 4 | 5HT2A activates junctional MyoII through Gβ13f/Gγ1, Dp114RhoGEF and Rho1. a** Schematic showing GPCR signaling modules that activate junctional and medial Rho1/MyoII. **b**–**e** Rho1 activity in 5HT2A loss-of-function (*SHT2A*−/−). Rho1-GTP snapshots in control (left panel) and *SHT2A*−/− (right panel) (**b**). Quantification of Rho-GTP sensor signal over-time in DV and AP oriented junctions (**c**), medial-apically (**d**), and amplitude of polarity (**e**) in different conditions. **f**–**h** Rho1 activity in 5HT2A overexpression (5HT2A + +). Images of Rho1-GTP at different time points (left to right) and different genotypes (top to bottom): control (top panel), 5HT2A + + (bottom panel) (**f**). Quantification of Rho1-GTP sensor signal in DV and AP oriented junctions (**g**), and amplitude of polarity (**h**) over-time. **i**–**l** Dp114RhoGEF levels in 5HT2A loss and gain-of-function. Snapshot of endogenous distribution of Dp114RhoGEF::mNeonGreen in control (left panel) and *SHT2A*−/− (right panel) (**i**); and quantification of signal in the respective genotypes (**j**). Snapshots of *sqh*-Dp114RhoGEF::GFP in control (left panel), and 5HT2A + +(right panel) (**k**); and quantification of the junctional signals (**l**) in the respective genotypes. **m**–**o** MyoII distribution in 5HT2A overexpression and Dp114RhoGEF knockdown (Dp114Rho-GEF-KD). Still images of MyoII in control (left), Dp114RhoGEF-KD (middle), and 5HT2A + + Dp114RhoGEF-KD (right) (**m**). Quantification of junctional MyoII intensities (**n**) and amplitude of polarity (**o**) for the above genotypes. **p**, **q** Gβ13f/Gγ1 overexpression (Gβ13f/Gγ1 + +) in *SHT2A*−/−. MyoII images in control (top-left), Gβ13f/Gγ1 + + (top-right), *SHT2A*−/− (bottom-left) and *SHT2A*-/- Gβ13f/Gγ1 + +

(bottom-right) (**p**). Quantification of junctional MyoII in different genotypes (**q**). In the box plots in (**c**, **d**, **e**, **g**, **h**, **j**, **l**, **n**, **o**, **q**) the line in the middle is plotted at the median. The box extends from the 25th to 75th percentiles. The whiskers indicate minimum and maximum values. Statistics: ns $P > 0.05$, *$P ≤ 0.05$, **$P ≤ 0.005$, ***$P ≤ 0.0005$ from two-tailed Mann–Whitney test. $P$ values in (**c**): 1 ($P = 0.026$), 2 ($P = 5.83 × 10^{-4}$), 3 ($P = 5.83 × 10^{-4}$), 4 ($P = 5.83 × 10^{-4}$), 5 ($P = 0.038$), 6 ($P = 5.83 × 10^{-4}$), 7 ($P = 5.83 × 10^{-4}$), 8 ($P = 5.83 × 10^{-4}$). $P$ values in (**d**): 1 ($P = 5.83 × 10^{-4}$), 2 ($P = 0.001$), 3 ($P = 5.83 × 10^{-4}$), 4 ($P = 5.83 × 10^{-4}$). $P$ values in (**e**): 1 ($P = 0.259$), 2 ($P = 0.456$), 3 ($P = 0.097$), 4 ($P = 0.456$). $P$ values in (**g**): 1 ($P = 0.129$), 2 ($P = 0.492$), 3 ($P = 0.007$), 4 ($P = 5.3 × 10^{-5}$), 5 ($P = 0.84$), 6 ($P = 0.206$), 7 ($P = 0.152$), 8 ($P = 1.06 × 10^{-4}$). $P$ values in (**h**): 1 ($P = 0.968$), 2 ($P = 0.351$), 3 ($P = 0.442$), 4 ($P = 0.272$). $P$ values in (**j**): 1 ($P = 0.02$), 2 ($P = 0.152$), 3 ($P = 0.003$), 4 ($P = 0.001$), 5 ($P = 0.001$), 6 ($P = 1.08 × 10^{-4}$). $P$ values in (**l**): 1 ($P = 0.004$), 2 ($P = 0.008$), 3 ($P = 0.004$), 4 ($P = 0.003$), 5 ($P = 0.126$), 6 ($P = 0.008$). $P$ values in (**n**): 1 ($P = 0.005$), 2 ($P = 0.018$), 3 ($P = 0.018$), 4 ($P = 0.005$), 5 ($P = 0.005$), 6 ($P = 0.003$), 7 ($P = 0.004$), 8 ($P = 0.004$), 9 ($P = 0.009$), 10 ($P = 0.004$), 11 ($P = 0.009$), 12 ($P = 0.004$). $P$ values in (**o**): 1 ($P = 0.003$), 2 ($P = 0.004$), 3 ($P = 0.945$). $P$ values in (**q**): 1 ($P = 7.99 × 10^{-4}$), 2 ($P = 0.018$), 3 ($P = 0.008$), 4 ($P = 0.008$), 5 ($P = 0.036$), 6 ($P = 0.005$), 7 ($P = 0.836$), 8 ($P = 0.181$), 9 ($P = 0.138$), 10 ($P = 0.295$), 11 ($P = 0.295$), 12 ($P = 0.005$), 13 ($P = 0.022$), 14 ($P = 0.056$), 15 ($P = 0.635$), 16 ($P = 0.313$), 17 ($P = 0.147$), 18 ($P = 0.635$), 19 ($P = 0.002$), 20 ($P = 0.002$), 21 ($P = 0.033$), 22 ($P = 0.033$), 23 ($P = 0.025$), 24 ($P = 0.001$). $n =$ number of embryos. Scale bars 5 µm. Source data are provided as a Source Data file.

embryos co-expressing the Vesicular Stomatitis Virus G-protein tagged with mNeonGreen (VsVg::mNeonGreen) and 5HT2B::mCherry, two proteins that are not expected to interact (negative control). All the constructs were expressed under the *sqh* promoter to ensure homogenous and similar expression levels. sFCCS analysis showed positive correlation for 5HT2A and 5HT2B (Fig. 5l, m), whereas no evidence of correlation was detected in the negative control. This argues that the two proteins physically interact in vivo and form heterocomplexes. Notably, the detected relative cross-correlation (-0.3 ± 0.1) is lower than the values (-0.4–0.8 depending on the choice of fluorophores) commonly detected on positive cross-correlation controls[61–63], i.e., for the case of 100% binding, indicating that the interaction of 5HT2A and 5HT2B is transient. By analyzing FCS amplitudes in each channel (see methods for details), a similar surface concentration of 5HT2A and 5HT2B was detected (Fig. 5n). Diffusion coefficients of -0.5 $\mu m^2/s$ were estimated for individual receptors in both the negative controls and in the co-expression condition (Fig. 5o), arguing that heterodimers will have similar diffusion dynamics as free receptors. Taken together, the genetic interaction, co-localization and cross-correlation analyses confirm a functional and physical interaction between 5HT2A and 5HT2B in vivo (Fig. 5p).

Based on this, we then asked how 5HT2B inhibits 5HT2A. We found that most of the 5HT2A::mNeonGreen co-localized with the dextran filled endocytic vesicles while most of the 5HT2B::mCherry cytoplasmic organelle did not (Supplementary Fig. 8a, a', b, b'). Consistently, 5HT2A::mNeonGreen cytoplasmic organelles were encapsulated in clathrin coated vesicles as assessed with the clathrin heavy chain ChC::RFPt (Supplementary Fig. 8d, d') and Rab5::GFP, which labels the early endosomes, did not colocalize with 5HT2B::mCherry cytoplasmic organelles (Supplementary Fig. 8e, e'). These observations suggest that 5HT2A undergoes endocytosis, while 5HT2B alone appears less likely to be internalized. Strikingly, when co-expressed, 5HT2A::mNeonGreen and 5HT2B::mCherry colocalize in the membrane and cytoplasmic organelles (Supplementary Fig. 8c, c'), consistent with the formation of heterocomplexes assessed by FCCS (Fig. 5l, m).

Based on this observation, we hypothesized that 5HT2B promotes 5HT2A endocytosis and compared the surface levels and the number of endocytic vesicles of 5HT2A::mNeonGreen in varying dosage of 5HT2B: 5HT2B mutant and endogenous levels (Supplementary Fig. 8f, f', f''). In the absence of 5HT2B, the 5HT2A::mNeonGreen signal at the membrane was higher compared to when expressed in the context of endogenous 5HT2B levels (Supplementary Fig. 8f, f'), consistent with the increased MyoII levels in the absence of 5HT2B (Fig. 5a–c and Supplementary Fig. 6a, b, d, e, f, h). Conversely, the number of 5HT2A::mNeonGreen cytoplasmic vesicles per cell was lower (Supplementary Fig. 8f, f''). Most of the cytoplasmic vesicles in the endogenous levels of 5HT2B were endocytic vesicles as they co-localized with the dextran filled mobile endocytic vesicles (Supplementary Fig. 8a, a'). This argues that 5HT2B promotes 5HT2A internalization. Collectively, 5HT2B inhibits 5HT2A signaling through heterodimerization and subsequent endocytosis. 5HT2A::mNeonGreen is recruited to the membrane in the absence of 5HT2B (Supplementary Fig. 8f, f'), whereas membrane levels are depleted in the absence of Toll receptors and accumulated in cytoplasmic organelles (Supplementary Fig. 4a, b), suggesting that Toll receptors are required to recruit or stabilize 5HT2A to the membrane and 5HT2B potentially regulates its membrane turnover/endocytosis.

### Serotonin receptors are required for tissue flows and MyoII activation during chick gastrulation

To investigate whether the activation of MyoII by the 5HT-receptors signaling during development is evolutionarily conserved in vertebrates, we studied it's function in the epiblast of gastrulating chick embryos (Fig. 6a). Serotonin has been detected in the chick embryo during the primitive streak formation[21]. Pharmacological treatments targeting the serotonergic system have been shown to affect chick gastrulation[19–21]. Contractile supracellular actomyosin cables are required to generate and integrate tissue scale deformations during chick gastrulation[34,35]. We examined the distribution of phosphorylated MyoII (phospho-MyoII) during the primitive streak formation following the treatment with the 5HT2A/2B antagonist Ritanserin (Fig. 6b and Supplementary Fig. 12). Embryos were treated with different concentrations of the inhibitor (20 μM, 100 μM, and 200 μM). Phospho-MyoII levels were markedly reduced in the treated embryos both in the contractile region and in the extra-embryonic tissue (EE); area opaca (Fig. 6b and Supplementary Fig. 12a–c). We observed a marked reduction in the phospho-MyoII signal at 20 μM and an even greater reduction at 200 μM (Fig. 6b). Both the junctional and medial phospho-MyoII levels were reduced (Fig. 6b and Supplementary Fig. 12a–c) while the amplitude of polarity was not affected in the treated embryos (Supplementary Fig. 12a–d). This suggests that 5HT2A/2B signaling is required for MyoII activation and regulation of MyoII levels but not polarization during chick gastrulation, consistent with that in *Drosophila*.

Tangential supracellular MyoII cables at the posterior margin drive oriented intercalation[34]. Fewer and shorter MyoII cables were observed in treated embryos (Fig. 6b and Supplementary Fig 12a). Consistent with this, we observed gastrulation defects such as: delayed primitive streak formation by 4–7 h at lower concentrations (50 μM, Supplementary Movie 20), contraction of the extra-embryonic (EE) and embryo proper (EP) tissues, whereas in the untreated embryos the EE tissue expands steadily and the EP tissue maintains a constant area[35] (Supplementary Movie 18). Primitive streak formation was absent in the embryos treated with 200 μM Ritanserin (Fig. 6c and Supplementary Movie 18). Large-scale vortex like tissue flows drive the primitive streak formation[34, 35]. In order to investigate whether these large-scale tissue flows were perturbed, we performed Particle Image Velocimetry (PIV) analysis[35] to follow the tissue motion in the brightfield movies. Consistently, no large-scale vortex like tissue flows were observed in the treated embryos (Fig. 6d and Supplementary Movie 19). Overall, we conclude that the role of serotonin receptors in activating actomyosin contractility and tissue flows during embryonic axis elongation is evolutionarily conserved in vertebrates.

## Discussion

We have reported that serotonin signaling has a morphogenetic function during embryo gastrulation through the regulation of actomyosin contractility in evolutionarily divergent lineages. We delineate two modes of regulation of MyoII activation by GPCRs in *Drosophila* (Supplementary Fig. 9): serotonin receptors control quantitatively Rho1, while planar polarization is controlled by Toll receptors together with Cirl/Latrophilin. Toll receptors encode polarity information and polarize Rho1/MyoII activity (Supplementary Fig. 10 and Supplementary Movie 17). Toll receptors such as Toll-8 interact with the adhesion GPCR Cirl/Latrophilin to form a self-organized polarizing module and recruit MyoII to the cell-cell interface[37]. The GPCR Cirl signaling contributes to both MyoII levels by regulating F-actin (Supplementary Fig. 11 and Supplementary Movie 10) and to junctional MyoII polarization by interacting with Toll receptors. 5HT2A requires Toll receptors for surface recruitment and MyoII hyperpolarization when overexpressed, and shows redundancy with Cirl to regulate F-actin (Supplementary Fig. 11 and Supplementary Movie 10), however, signals independently of Cirl to form a distinct signaling module to regulate MyoII levels. 5HT2A/5HT2B physically interact to form heterocomplexes and regulate MyoII levels, forming a module that transduces quantitative information. Our data show that 5HT2B regulates the 5HT2A signaling and membrane levels through heterodimerization and endocytosis. Another GPCR, Smog, regulates the level of junctional Rho1/MyoII activation and signals independent of

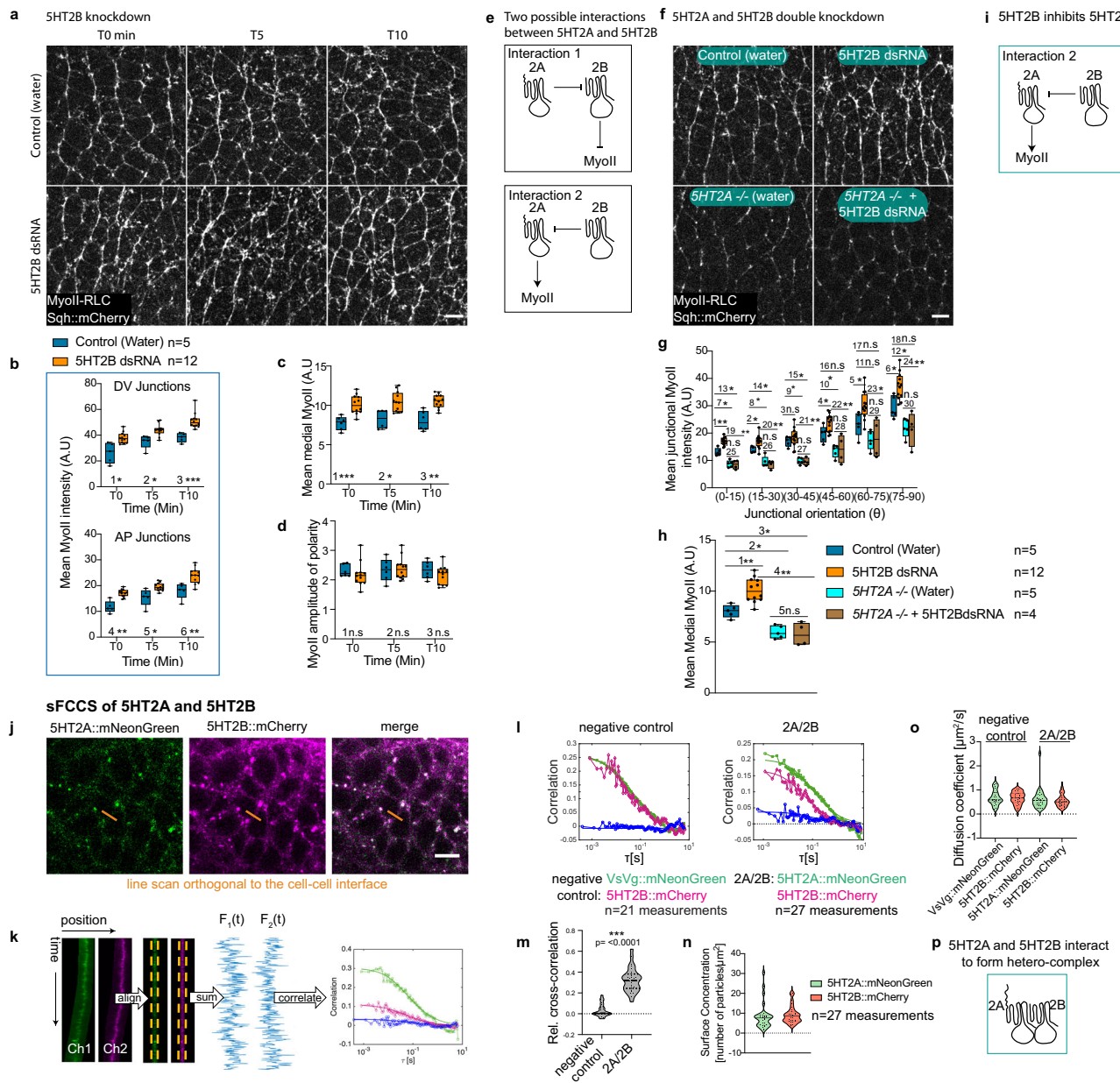

Fog at the junctions[38]. Collectively, we conclude that multiple GPCRs are required to regulate MyoII levels. There is no single ligand that encodes quantitative information. Multiple GPCRs interpret specific ligand(s) and the ligand-receptor stoichiometry determined by homodimerization[64]/heterodimerization and subsequent endocytosis, encode and integrate quantitative information.

We found that the serotonin/receptors signaling also activates MyoII contractility and regulates MyoII quantity but not polarity during chick gastrulation. How MyoII is activated and polarized during primitive streak formation in the chick is unknown. GPCR signaling instructs the activation and polarization of MyoII during axis extension in *Drosophila*[37–39]. By studying two phylogenetically distant species (separated by ~250 million years), we uncovered a common morphogenetic function of serotonin and GPCR serotonin receptors in modulating cellular mechanics. Further studies in a wider range of organisms are needed to understand the evolutionary history of the mechanical function of serotonin signaling. Our finding opens a new door for the investigation of the serotonergic system in other organisms, such as sea urchin, zebrafish, *Xenopus* and mouse[2,17–26,65], reviewed in ref. 14, where the serotonergic system has been detected in

early embryos and morphogenetic function has been documented in some organisms, but the mechanism remains elusive. Serotonin receptors and transporters have been detected in the human oocyte and in embryos four days after fertilization[66,67]. These reports suggest a potential role for the serotonergic system in human development. Cellular mechanics plays a major role in maintaining tissue homeostasis and integrity during development and in the adult. Indeed, serotonin and serotonin receptors are expressed in a wide range of organs. Inhibition of serotonin receptors results in heart defects and incomplete neural tube closure in mouse[26] and chicken[19]. Neural tube closure requires actomyosin contractility[68,69]. Our findings highlight the need to investigate the role of the serotonergic system beyond its classical physiological function in the nervous system.

GPCR signaling is involved in the regulation of actomyosin contractility during morphogenesis in several organisms. For example, Fog/Smog/Mist and Toll/Cirl in *Drosophila*[37,38,41,51], Wnt/Frzzled in *C. elegans*[70]. Serotonin/5HT2A/5HT2B in *Drosophila* and chick, as we report in this study. All the metazoan G *alpha* family, including Gα$_{12/13}$, which is known to regulate actomyosin contractility during gastrulation in *Drosophila*[38,71,72] and required for gastrulation in zebrafish[73], are

**Fig. 5 | 5HT2B represses MyoII accumulation in the ectoderm and requires 5HT2A. a–d** MyoII distribution in 5HT2B knockdown by RNAi (5HT2B dsRNA) injection. Images of MyoII for different time points (left to right); in water injected control (top panels) and 5HT2B dsRNA injection (bottom panels) (**a**). Quantification of time-traces of MyoII in DV and AP oriented junctions (**b**), medial MyoII (**c**), and amplitude of polarity (**d**). **e** Schematic depicting two possible interactions between 5HT2A and 5HT2B. **f–h** Genetic interaction between 5HT2A and 5HT2B by knocking down 5HT2B in *SHT2A−/−*. MyoII images for water injected control (top-left), 5HT2B dsRNA (top-right), water injected *SHT2A−/−* (bottom-left) and *SHT2A−/−* injected with 5HT2B dsRNA (bottom-right) (**f**). Quantification of junctional (**g**), and medial (**h**) MyoII intensities for different conditions. **i** Schematic showing the 5HT2B inhibition of 5HT2A as supported by the genetic interaction above. **j–p** Scanning FCCS (sFCCS) of 5HT2A and 5HT2B. **j** Confocal microscopy images of cells co-expressing 5HT2A::mNeonGreen and 5HT2B::mCherry. sFCCS measurements were performed perpendicular to the plasma membrane, as shown (orange line). **k** Schematic of sFCCS. Line scans acquired sequentially in line interleaved excitation mode were aligned computationally to correct for lateral cell or tissue movement. Membrane pixels were identified and integrated to provide membrane fluorescence time series in each channel. Autocorrelation functions (ACFs, green/magenta) in each channel and the cross-correlation function (CCF, blue) between two channels were calculated (see 'Methods' for details). **l** Representative correlation functions obtained in a negative control sample consisting of a line co-expressing VsVg::mNeonGreen and 5HT2B::mCherry (left) and in cells co-expressing the two receptors 5HT2A::mNeonGreen and 5HT2B::mCherry (2A/2B) (right). Green= ACF of the mNeonGreen channel, magenta=ACF of the mCherry channel and blue=CCF. In the negative control, the CCF (blue) fluctuates around zero, indicating absence of co-diffusion and hence hetero-interactions. In the 2A/2B sample, a positive CCF (blue) is obtained, indicating a co-diffusing species, i.e., the presence of heterocomplexes. *n* = number of independent measurements (number of cell-cell interfaces scanned) from 6 negative control embryos and 7 embryos co-expressing 5HT2A::mNeonGreen and 5HT2B::mCherry. Each embryo was considered as an independent experiment. **m** Quantification of the relative cross-correlation in negative control and 5HT2A::mNeonGreen and 5HT2B::mCherry (2A/2B) co-expressing embryos. Statistical significance was assessed by (two-tailed) Mann−Whitney test (***$P \leq 0.0005$). **n** Quantification of surface concentration of 5HT2A::mNeonGreen and 5HT2B::mCherry. **o** Quantification of diffusion coefficient in different conditions. **p** Schematic showing the interaction between 5HT2A and 5HT2B. In the violin plot in (**m**–**o**), the thick line in the middle is the median, the dotted lines are the interquartile range. In the box plots in (**b**–**d**, **g**, **h**) the line in the middle is plotted at the median. The box extends from the 25th to 75th percentiles. The whiskers indicate minimum and maximum values. Statistics: ns $P > 0.05$, *$P \leq 0.05$, **$P \leq 0.005$, ***$P \leq 0.0005$ from two-tailed Mann−Whitney test. *P* values in (**b**): 1 ($P = 0.006$), 2 ($P = 0.006$), 3 ($P = 3.23 \times 10^{-4}$), 4 ($P = 0.001$), 5 ($P = 0.009$), 6 ($P = 0.002$). *P* values in (**c**): 1 ($P = 4.31 \times 10^{-4}$), 2 ($P = 0.024$), 3 ($P = 7.5 \times 10^{-4}$). *P* values in (**d**): 1 ($P = 0.384$), 2 ($P = 0.892$), 3 ($P = 0.291$). *P* values in (**g**): 1 ($P = 0.001$), 2 ($P = 0.019$), 3 ($P = 0.279$), 4 ($P = 0.037$), 5 ($P = 0.019$), 6 ($P = 0.019$), 7 ($P = 0.008$), 8 ($P = 0.008$), 9 ($P = 0.008$), 10 ($P = 0.008$), 11 ($P = 0.056$), 12 ($P = 0.016$), 13 ($P = 0.016$), 14 ($P = 0.016$), 15 ($P = 0.016$), 16 ($P = 0.111$), 17 ($P = 0.286$), 18 ($P = 0.111$), 19 ($P = 0.001$), 20 ($P = 0.001$), 21 ($P = 0.001$), 22 ($P = 0.004$), 23 ($P = 0.008$), 24 ($P = 0.001$), 25 ($P = 0.905$), 26 ($P > 0.999$), 27 ($P = 0.905$), 28 ($P = 0.905$), 29 ($P = 0.905$), 30 ($P > 0.999$). *P* values in (**h**): 1 ($P = 0.001$), 2 ($P = 0.008$), 3 ($P = 0.016$), 4 ($P = 0.001$), 5 ($P = 0.73$). *n* = number of embryos. For sFCCS, *n* = number of independent measurements. Scale bars 5 μm. Source data are provided as a Source Data file.

conserved in the common metazoan ancestor[74]. Our study highlights the evolutionarily conserved function of GPCR signaling in regulating actomyosin contractility and cellular mechanics during morphogenesis.

The presence of serotonin precedes the appearance of the nervous system on both evolutionary and developmental time scales. Serotonin is present in unicellular eukaryotes that evolved more than 600 million years ago. For example, *Tetrahymena*, can produce and respond to serotonin and receptor inhibitor, that alters phagocytosis and ciliary motion, which requires actin-dependent contractility[5,75,76]. Serotonin has been detected in primitive nerveless metazoans: sponges[77,78] that evolved nearly 500–600 million years ago[79]; and has been shown to induce coordinated contractions[80] that require actomyosin[81]. Insects evolved around ~440 million years ago, flying insects after 40 million years[82], birds evolved around 150 million years ago[83]. By comparing the two species whose common ancestor lived 250 million years ago, we show the functional conservation of serotonin/receptors signaling in the regulation of actomyosin contractility during embryogenesis. Later, in the adult organisms, the serotonin signaling predominantly modulates organismal behavior. This provides strong evidence for a strategy implemented by evolution to repurpose a cellular mechanism contextually, without having to innovate a new mechanism completely from scratch. The serotonin signaling might have evolved primarily to regulate intracellular signaling, cellular contractility, and collective cell behavior and later evolved in the nervous system for inter-cellular communication.

## Methods
### Fly strains and genetics
MyoII dynamics were visualized with *sqh-Sqh::mCherry* (the regulatory light chain of MyoII, sqh, fused to mCherry, under the sqh promoter) except for Fig. 3h–k and Supplementary Fig. 7e–g, sqh-Sqh::GFP lines were used. *SHT2A-GAL4, SHT2B-GAL4* (refs. 43,44) (were gift from Leslie B. Vosshall, Rockefeller University, USA and Yi Rao, Peking University, China). *UAS-SHT2A* (BDSC#24504), *UAS-Trh* (BDSC#27638 and BDSC#27639), *Trh[01]*(BDSC#86147; refs. 44). HT2B RNAi (BDSC # 60488), *sqh-SHT2A::mNeonGreen* and *sqh-SHT2B::mCherry* (generated in the laboratory). *Endo-Dp114RhoGEF::mNeonGreen* (generated in the lab), *67GAL4,sqh-Dp114RhoGEF::GFP* (ref. 39), *ANI-RBD::mEGFP* (ref. 54), *ANI-RBD::mNeonGreen,sqhCherry* (generated in the lab), UAS-Gβ13f and UAS-Gγ1 (ref. 39), UAS-5HT2B (gift from David Krantz Lab UCLA, USA; ref. 44). Homozygous *Ecad::GFP,sqh-Sqh::mCherry* line was used to visualize MyoII in 5HT2A loss-of-function (Fig.1b–g), Trh loss-of-function (Fig. 2b–d), 5HT2B loss-of-function (Supplementary Fig. 6a–d), 5HT2B knock-down (Fig. 5a–d) and 5HT2A and 5HT2B double knock-down (Fig. 5f–h) and therefore have two copies of *sqh-Sqh::mCherry*. All the overexpression experiments to visualize MyoII have a single copy of *sqh-Sqh::mCherry* or *sqh-Sqh::GFP* and were performed using the *67GAL4,Ecad::GFP; sqh-Sqh::mCherry* line (ref. 39) except for Fig. 3h–k where the *67GAL4;sqh-Sqh::GFP,Lifeact::mCherry* line was used. MyoII levels are higher in the two copy controls compared to the single copy controls. All the genetic crosses are detailed below.

### Fly genetics by figures
Figure 1: **b–g** Control: *Ecad::GFP,sqh-Sqh::mCherry,/;/Ecad::GFP,sqh-Sqh::mCherry*; (males and females); *SHT2A-/-: SHT2A-GAL4* null mutant on third chromosome combined with *Ecad::GFP,sqh-Sqh::mCherry* on the second chromosome. Embryos obtained from the homozygous (*Ecad::GFP,sqh-Sqh::mCherry/Ecad::GFP,sqh-Sqh::mCherry;5HT2A-GAL4/5HT2A-GAL4*) males and females were imaged. **h–m** Control: *67GAL4,Ecad::GFP,sqh-Sqh::mCherry/+* (females) crossed with *yw* (males); 5HT2A + +: *;67GAL4,Ecad::GFP,sqh-Sqh::mCherry/UAS-5HT2A*; (females) crossed with homozygous;*UAS-SHT2A/UAS-SHT2A*; (males).

Figure 2: **b–d** Control: *Ecad::GFP,sqh-Sqh::mCherry/Ecad::GFP,sqh-Sqh::mCherry*; (males and females); *Trh[01]: Trh[01]* null mutant on the third chromosome combined with;*Ecad::GFP,sqh-Sqh::mCherry*; on the second chromosome. Embryos obtained from the homozygous (;*Ecad::GFP,sqh-Sqh::mCherry/Ecad::GFP,sqh-Sqh::mCherry; Trh[01]/Trh[01]*) males and females were imaged. **e–h** Control: *67GAL4,Ecad::GFP,sqh-Sqh::mCherry/+* (females) crossed with *yw* (males); Trh + +: *;67GAL4,Ecad::GFP,sqh-Sqh::mCherry/UAS-Trh*; (females) crossed with homozygous;*UAS-Trh/UAS-Trh*; (males). **i–k** Control: *;67GAL4,Ecad::GFP,sqh-Sqh::mCherry/+* (females) crossed with *yw* (males); Trh + +: *;67GAL4,Ecad::GFP,sqh-Sqh::mCherry/UAS-Trh*; (females) crossed with homozygous;*UAS-Trh/UAS-Trh*; (males). *SHT2A-/-: ;67GAL4,Ecad::GFP,sqh-Sqh::*

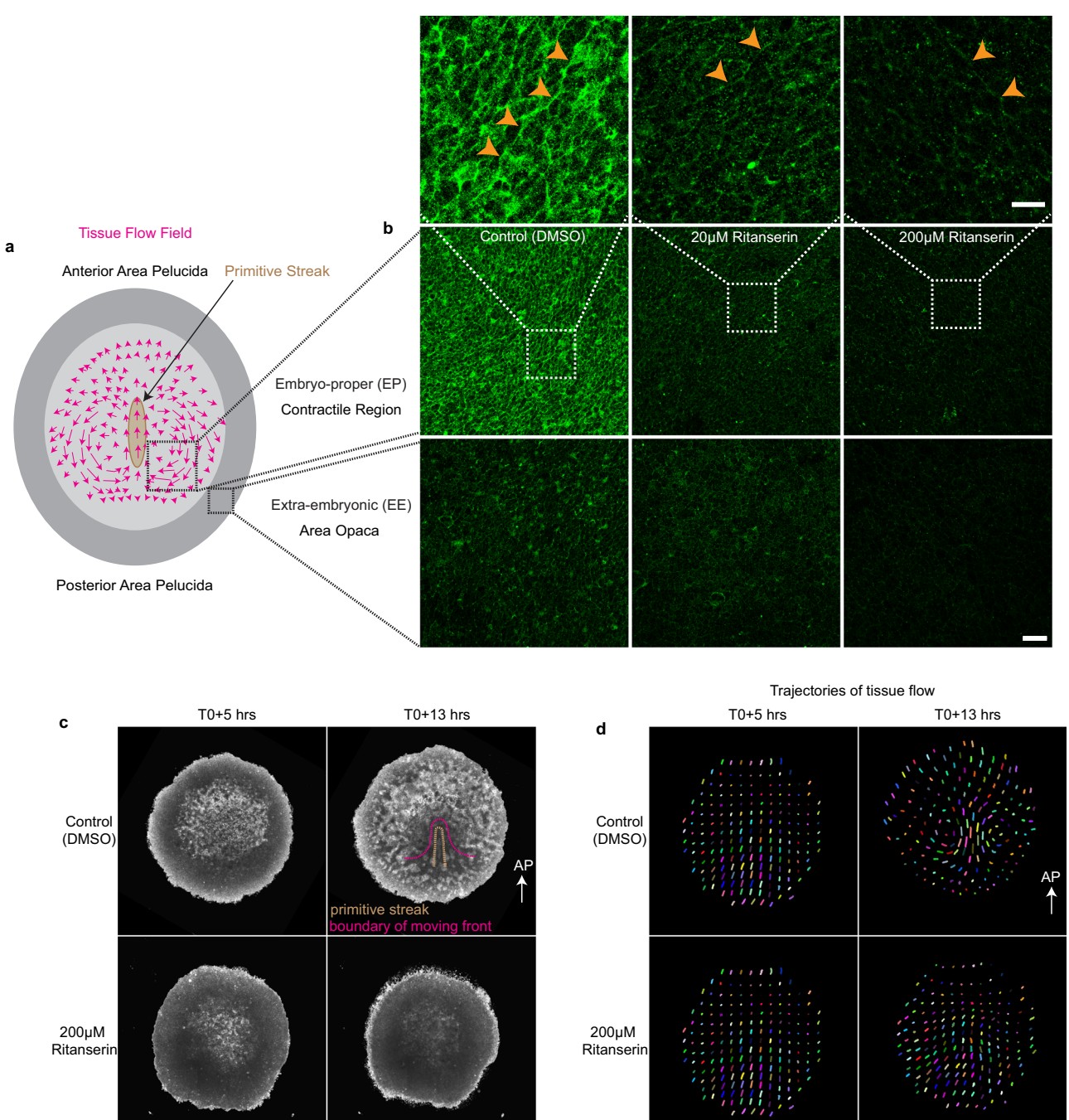

**Fig. 6 | Effect of serotonin receptor inhibition on tissue flows and myosin activity during chick gastrulation. a** Schematic of the gastrulating chick embryo. **b** Representative images of phospho-MyoII in the embryo-proper (EP), contractile region (top panels) and extra-embryonic tissue (EE), area opaca (bottom panels) in the DMSO treated control (left), 20 μM (middle) and 200 μM (right) Ritanserin treated embryos. The white box in the images is magnified in the panels above each condition. The orange arrowheads indicate supracellular MyoII cables along the aligned cell junctions. The scale bar in the bottom left panel is 50 μm, in the inset 20 μm. **c** Snapshot of the brightfield images taken from the Video 18 at T0 + 5 h (left) and T0 + 13 h (right). Control treated with DMSO (top panels) and Ritanserin 200 μM (bottom panels). The scale bar is 500 μm. **d** Particle Image Velocimetry (PIV) images of the embryos in Video 18 at T0 + 5 h (left) and T0 + 13 h (right). The AP arrow in (**c**, **d**) indicates the direction of anterior posterior axis. The pink dotted line in **c** is the boundary of the tissue flow traced from the PIV image of the corresponding time in (**d**). Images in (**b**, **c**) are representative of at least three embryos per condition, experiments were replicated at least three times.

*Cherry/+;SHT2A-GAL4/SHT2A-GAL4* (females) crossed with;*;SHT2A-GAL4/SHT2A-GAL4* (males). *SHT2A-/-* Trh + +: *;67GAL4,Ecad::GFP,sqh-Sqh::mCherry/UAS-Trh;SHT2A-GAL4/SHT2A-GAL4* (females) crossed with homozygous;*UAS-Trh/UAS-Trh; SHT2A-GAL4/SHT2A-GAL4* (males). **l–o** Control: *;67GAL4,Ecad::GFP,sqh-Sqh::mCherry/+* (females) crossed with *yw* (males); 5HT2A + +: *;67GAL4,Ecad::GFP,sqh-Sqh::mCherry/UAS-SHT2A;* (females) crossed with;*UAS-5HT2A/ UAS-5HT2A;* (males).

*Trh[01]* 5HT2A + +: *;67GAL4,Ecad::GFP,sqh-Sqh::mCherry/UAS-SHT2A; Trh[01]/Trh[01]* (females) crossed with;*UAS-SHT2A/UAS-SHT2A; Trh[01]/Trh[01]* (males).

Figure 3: **b–d** WT (water): *67GAL4,Ecad::GFP,sqh-Sqh::mCherry/+* (females) crossed with *yw* (males); 5HT2A + + (water): *;67GAL4,Ecad::GFP,sqh-Sqh::mCherry/UAS-SHT2A;* (females) crossed with;*UAS-SHT2A/ UAS-SHT2A;* (males). WT + Toll2,6,8 RNAi: *;67GAL4,Ecad::GFP,sqh-Sqh::mCherry/+* (females) crossed with *yw* (males); 5HT2A + + Toll2,6,8

RNAi: ;67GAL4,Ecad::GFP,sqh-Sqh::mCherry/UAS-5HT2A; (females) crossed with;UAS-5HT2A/ UAS-5HT2A; (males). **e–g** Control:;sqh-Sqh::mCherry/+; (females) crossed with yw (males). Cirl-/-;sqh-Sqh::mCherry, Cirl^{KO}/Cirl^{KO}; (females) crossed with;Cirl^{KO}/Cirl^{KO}; (males). 5HT2A-/-;sqh-Sqh::mCherry/+; 5HT2A-GAL4/5HT2A-GAL4 (females) crossed with;; 5HT2A-GAL4/5HT2A-GAL4 (males). Cirl-/+ 5HT2A-/+:;sqh-Sqh::mCherry, Cirl^{KO}/+; 5HT2A-GAL4/+ (females) crossed with yw (males). Cirl-/+ 5HT2A-/-;sqh-Sqh::mCherry, Cirl^{KO}/+; 5HT2A-GAL4/5HT2A-GAL4 (females) crossed with;;5HT2A-GAL4/5HT2A-GAL4 (males). Cirl-/- 5HT2A-/+:;sqh-Sqh::mCherry, Cirl^{KO}/ Cirl^{KO}; 5HT2A-GAL4/+ (females) crossed with;Cirl^{KO}/Cirl^{KO}; (males). **h–k** Control: 67GAL4/+;sqh-Sqh::GFP,Lifeact::mCherry/+ (females) crossed with ;67GAL4/ 67GAL4; (males). Cirl -/-;Cirl^{KO}/67GAL4,Cirl^{KO}; sqh-Sqh::GFP,Lifeact::mCherry/+ (females) crossed with ;67GAL4,Cirl^{KO}/67GAL4,Cirl^{KO}; (males). Cirl-/- 5HT2A + +: ;67GAL4,Cirl^{KO}/67GAL4,Cirl^{KO}; sqh-Sqh::GFP,Lifeact::mCherry/UAS-5HT2A (females) crossed with; 67GAL4,Cirl^{KO}/67GAL4,Cirl^{KO}; UAS-5HT2A/UAS-5HT2A (males).

Figure 4: **b–e** Control: UBI-ANI-RBD::mEGFP/+; females and yw males. SHT2A-/-;;UBI-ANI-RBD::mEGFP/+; 5HT2A-GAL4/5HT2A-GAL4 females crossed with;; 5HT2A-GAL4/5HT2A-GAL4 males. **f–h** Control: ;67GAL4/ + ; UBI-ANI-RBD::mEGFP/+ (females) crossed with yw (males). 5HT2A + +: ;67GAL4/UAS-5HT2A; UBI-ANI-RBD::mEGFP/+ (females) crossed with;UAS-5HT2A/UAS-5HT2A; (males). **i, j** Control: EndoDp114RhoGEF::mNeonGreen/EndoDp114RhoGEF::mNeonGreen; (males crossed with females). SHT2A-/-;;EndoDp114RhoGEF::mNeonGreen;/EndoDp114RhoGEF::mNeonGreen;5HT2A-GAL4/5HT2A-GAL4 (males crossed with females). **k–l** Control: 67GAL4/+; sqh-eGFP::Dp114RhoGEF/+ (females) crossed with yw (males). 5HT2A + +: ;67GAL4/UAS-5HT2A; sqh-eGFP::Dp114RhoGEF/+ (females) crossed with;UAS-5HT2A/UAS-5HT2A; (males). **m–o** Control: 67GAL4,EcadGFP,sqh-sqh::mCherry/+ (females) crossed with yw (males); Dp114RhoGEF-KD: ;67GAL4,Ecad::GFP,sqh-Sqh::mCherry/UAS-Dp114RhoGEF-shRNA (females) crossed with;UAS-Dp114RhoGEF-shRNA/ UAS-Dp114RhoGEF-shRNA; (males); Dp114 RhoGEF-KD 5HT2A + +: ;67GAL4,Ecad::GFP,sqh-Sqh::mCherry/UAS-Dp114RhoGEF-shRNA;UAS-5HT2A/+ (females) crossed with;UAS-Dp114RhoGEF-shRNA/ UAS-Dp114RhoGEF-shRNA; UAS-5HT2A/UAS-5HT2A (males); **p–q** Control: 67GAL4,Ecad::GFP,sqh-Sqh::mCherry/+ (females) crossed with yw (males); SHT2A-/-: ;67GAL4,Ecad::GFP,sqh-Sqh::mCherry/ +;5HT2A-GAL4/5HT2A-GAL4 (females) crossed with;;5HT2A-GAL4/5HT2A-GAL4 (males). Gβ13f Gγ1 + +: ;67GAL4,Ecad::GFP,sqh-Sqh::mCherry/UAS-Gβ13f;UAS-Gγ1/+ (females) crossed with;UAS-Gβ13f/ UAS-Gβ13f;UAS-Gγ1/UAS-Gγ1 (males). SHT2A-/- Gβ13f Gγ1 + +: ;67GAL4,Ecad::GFP,sqh-Sqh::mCherry/UAS-Gβ13f;5HT2A-GAL4,UAS-Gγ1/5HT2A-GAL4,UAS-Gγ1 (females) crossed with;UAS-Gβ13f/UAS- Gβ13f;5HT2A-GAL4,UAS-Gγ1/5HT2A-GAL4,UAS-Gγ1 (males).

Figure 5: **a–d** Control: Ecad::GFP,sqh-Sqh::mCherry/Ecad::GFP,sqh-Sqh::mCherry; males and females. 5HT2B dsRNA:;Ecad::GFP,sqh-Sqh::mCherry/Ecad::GFP,sqh-Sqh::mCherry; males and females. **f–h** Control: Ecad::GFP,sqh-Sqh::mCherry/Ecad::GFP,sqh-Sqh::mCherry; males and females. 5HT2B dsRNA:;Ecad::GFP,sqh-Sqh::mCherry/Ecad::GFP,sqh-Sqh::mCherry; males and females. SHT2A-/-;;Ecad::GFP,sqh-Sqh::mCherry/Ecad::GFP,sqh-Sqh::mCherry; 5HT2A-GAL4/5HT2A-GAL4 males and females. SHT2A-/- + 5HT2BdsRNA:;Ecad::GFP,sqh-Sqh::mCherry/Ecad::GFP,sqh-Sqh::mCherry; 5HT2A-GAL4/5HT2A-GAL4 males and females. **j–o** negative control: sqh-VsVg::mNeonGreen/sqh-5HT2B::mCherry (males and females). 2A/2B:;sqh-SHT2A::mNeonGreen/sqh-SHT2A::mNeonGreen; sqh-SHT2B::mCherry/sqh-SHT2B::mCherry (males and females).

Supplementary Fig. 1: **a, b** WT: yw males and females. SHT2A-/-:;;SHT2A-GAL4/SHT2A-GAL4 males and females. SHT2B-/-:;;SHT2B-GAL4/5HT2B-GAL4 males and females.

Supplementary Fig. 2: **a–c, e–g** Control: Ecad::GFP,sqh-Sqh::mCherry;/;Ecad::GFP,sqh-Sqh::mCherry; (males and females). SHT2A-/-: (Ecad::GFP,sqh-Sqh::mCherry/Ecad::GFP,sqh-Sqh::mCherry; SHT2A-GAL4/SHT2A-GAL4. **d, h–k** Control: 67GAL4,Ecad::GFP,sqh-Sqh:: mCherry/+ (females) crossed with yw (males); 5HT2A + +: ;67GAL4,E-cad::GFP,sqh-Sqh::mCherry/UAS-5HT2A; (females) crossed with homozygous;UAS-5HT2A/UAS-5HT2A; (males).

Supplementary Fig. 3: **b–d** Control: sqh-Sqh::mCherry/sqh-Sqh::mCherry males and females. sqh-5HT2A::neonGreen:; sqh-5HT2A::mNeonGreen/ sqh-5HT2A::mNeonGreen;sqh-Sqh::mCherry/sqh-Sqh::mCherry males and females. **e, f** sqh-5HT2A::mNeonGreen/ sqh-5HT2A::mNeonGreen; sqh-GAP43::mCherry/sqh-GAP43::mCherry (males and females).

Supplementary Fig. 4: sqh-5HT2A::mNeonGreen/sqh-5HT2A::mNeonGreen; sqh-Sqh::mCherry/ sqh-Sqh::mCherry.

Supplementary Fig. 5: **a, b** Control: UAS-MBS::GFP/67GAL4; (females) crossed with ;67GAL4/67GAL4; (males). 5HT2A + +:;UAS-MBS::GFP/67GAL4;UAS-5HT2A/ + (females) crossed with ;67GAL4/ 67GAL4; (males). **c** sqh-5HT2A::mNeonGreen/ sqh-5HT2A::mNeonGreen; sqh-Sqh::mCherry/ sqh-Sqh::mCherry.

Supplementary Fig. 6: **a–d** Control: Ecad::GFP,sqh-Sqh::mCherry,/ ;Ecad::GFP,sqh-Sqh::mCherry; (males and females); SHT2B-/-: SHT2B-GAL4 null mutant on third chromosome combined with Ecad::GFP,sqh-Sqh::mCherry on the second chromosome. Embryos obtained from the (;Ecad::GFP,sqh-Sqh::mCherry/Ecad::GFP,sqh-Sqh::mCherry;5HT2B-GAL4/TM6B,hum) males and females were imaged. We did not observe homozygous adults when 5HT2B-GAL4 was combined with;-Ecad::GFP,sqh-Sqh::mCherry/Ecad::GFP,sqh-Sqh::mCherry; line. **e–h** Control: 67GAL4,Ecad::GFP,sqh-Sqh::mCherry/+ (females) crossed with yw (males); 5HT2BshRNA: ;67GAL4,Ecad::GFP,sqh-Sqh::mCherry/UAS-5HT2B shRNA; (females) crossed with homozygous;UAS-5HT2B-shRNA/UAS-5HT2B-shRNA; (males).

Supplementary Fig. 7: **b–d** Control: 67GAL4,Ecad::GFP,sqh-Sqh::mCherry/+ (females) crossed with yw (males); 5HT2B-WT + +: ;67GAL4,Ecad::GFP,sqh-Sqh::mCherry/UAS-5HT2B-WT; (females) crossed with homozygous;UAS-5HT2B-WT/UAS-5HT2B-WT; (males). **e–g** Control: sqh-Sqh::GFP/sqh-Sqh::GFP; (males and females). 5HT2B::mCherry + +:;sqh-Sqh::GFP/sqh-Sqh::GFP; sqh-SHT2B::mCherry/ sqh-SHT2B::mCherry (males and females). **h–i** sqh-SHT2B::mCherry/ sqh-SHT2B::mCherry; (males and females).

Supplementary Fig. 8: **a, a′** sqh-SHT2A::mNeonGreen/ sqh-SHT2A::mNeonGreen; (males and females). **b, b′** sqh-5HT2B::mCherry/ sqh-5HT2B::mCherry (males and females). **c, c′** sqh-5HT2A::mNeonGreen/ sqh-5HT2A::mNeonGreen; sqh-5HT2B::mCherry/ sqh-SHT2B::mCherry (males and females). **d, d′** sqh-SHT2A::mNeonGreen/ChC::RFPt; (males and females). **e, e′** ;sqh-5HT2B::mCherry/sqh-SHT2B::mCherry;Rab5::GFP/Rab5::GFP (males and females). **f, f′, f″** SHT2B -/-:;sqh-SHT2A::mNeonGreen/sqh-SHT2A::mNeonGreen; SHT2B-GAL4/TM6C,sb (males and females). Endo 5HT2B:;sqh-5HT2A::mNeonGreen/sqh-SHT2A::mNeonGreen.

Supplementary Fig. 10: Control: ;;ANI-RBD::mNeonGreen,sqh-Sqh::mCherry/ ANI-RBD::mNeonGreen,sqh-Sqh::mCherry. Toll 2,6,8 RNAi:;;;ANI-RBD::mNeonGreen,sqh-Sqh::mCherry/ ANI-RBD::mNeonGreen,sqh-Sqh::mCherry.

Supplementary Fig. 11: Control: 67GAL4/+;sqh-Sqh::GFP,Lifeact::mCherry/+ (females) crossed with ;67GAL4/ 67GAL4; (males). Cirl -/-;Cirl^{KO}/67GAL4,Cirl^{KO}; sqh-Sqh::GFP,Lifeact::mCherry/+ (females) crossed with ;67GAL4,Cirl^{KO}/67GAL4,Cirl^{KO}; (males). Cirl-/- 5HT2A + +: ;67GAL4,Cirl^{KO}/67GAL4,Cirl^{KO}; sqh-Sqh::GFP,Lifeact::mCherry/UAS-5HT2A (females) crossed with ;67GAL4,Cirl^{KO}/67GAL4,Cirl^{KO}; UAS-5HT2A/UAS-5HT2A (males).

## Constructs and transgenesis

Endo-mNeonGreen::Dp114RhoGEF: In order to tag Dp114RhoGEF N-terminally with mNeonGreen at its locus, a KO-attP founder line was first generated by CRISPR/Cas9 mediated editing of the gene (editing performed by InDroso, Christine Le Borgne in Rennes). Whole ORF and UTRs were deleted (from 22 bases after the first non-coding exon to 81 bases before the last exon end) and replaced by an attP-LoxP cassette.

This "KO-attP founder" line was sent to BestGene Inc and used for PhiC31 mediated transgenesis. A plasmid containing the deleted genomic sequence plus an N-ter mNeonGreen tag was built, resulting in the N-ter tagging of the gene at the locus. Detailed strategies and sequences are available upon request. Flies are homozygous viable.

UBI-mNeonGreen::ANILrbd (ANI-RBD::mNeonGreen): To construct the mNeonGreen-tagged Rho1-GTP sensor, the same strategy was used as for the mEGFP::ANI-RBD Rho sensor (Munjal et al.[54]). mNeonGreen ORF was fused to the C-terminal end of Drosophila anillin (amino acids 748–1239, Genebank ID: AAL39665,), which lacks the N-terminal myosin and actin binding domain but retains its Rho1 binding domain. A GGSGGGSGGGS flexible linker was inserted between mNeonGreen and anillin. The transgene is expressed under the ubiquitin p63E promoter. For cloning, the mNeon-Green::ANILrbd ORF replaced the stop-CD8GFP ORF in the Ubi-stop-mCD8GFP plasmid obtained from Stepan Belyakin, Russia (Gene-Bank ID: KC845568).

UASt-5HT2A::Wt and mNeonGreen: ORF corresponding to 5HT2A-RB was amplified (using RE48265 EST clone) and cloned into UASt-attB. Cter mNeonGreen tag was added after a GGSGGGS flexible aa linker.

sqh-5HT2A-mNeonGreen: The ORF corresponding to 5HT2A-RB was amplified (using the RE48265 EST clone) and cloned into the modified sqh promoter (Garcia et al.[39]), replacing the ORF of sqh-mCherry. Cter mNeonGreen tag was added after a GGSGGGS flexible aa linker.

sqh-5HT2B-mCherry: The ORF corresponding to 5HT2B was amplified and cloned into a modified sqh promoter (Garcia et al.[39]), replacing the ORF of sqh-mCherry. The Cter mCherry tag was added after a GGSGGGS flexible aa linker.

All recombinant expression vectors were constructed using in-fusion cloning (Takara Bio), verified by sequencing (Genewiz) and sent to BestGene Incorporate for PhiC31 site-specific mediated transgenesis in both 9736 (2R, 53B2) and attP2 (3L, 68A4). Fully annotated FASTA sequences of all these vectors and detailed cloning strategies are available on request.

## RNA interference in embryos

**dsRNA probes.** 5HT2B dsRNA: A 544-bp long dsRNA probes against 5HT2B (CG42796; https://www.ncbi.nlm.nih.gov/gene/41017) were constructed encompassing ATG (312b before and 229b after). A PCR product containing the sequence of the T7 promoter (TAA-TACGACTCACTATAGG) followed by 22–24 nucleotides specific for the gene (forward primer: AATGTTTGCCACCAATATCCGTTC; reverse primer: GGGCCCAGTAGTTGTTCGCATC) was gel purified and then used as a template for in vitro RNA synthesis with T7 polymerase using HiScribe™ T7 Quick High Yield RNA Synthesis Kit (NEB, E0250). The dsRNA probe was purified using Sure-Clean (Bioline, BIO-37047) and diluted to a final concentration of 5 µM in RNAse-free water prior to injection. The complete sequence of the forward (T7-5HT2B-F1) and reverse (T7-5HT2B-R1) primers and dsRNA probe is provided below.

T7-5HT2B-F1:TAATACGACTCACTATAGGAATGTTTGCCACCAAT ATCCGTTC

T7-5HT2B-R1:TAATACGACTCACTATAGGGGGCCCAGTAGTTGT TCGCATC

>5HT2B-FR1, 543 bp; 312 bp upstream ATG to aa A77

AATGTTTGCCACCAATATCCGTTCGGTTGCCGTAGAAATCGTCC TGGTGGCGCCGTGAGCTCTCGTCCTTTTCCTTTCGCGAGTCCTGAAT CCCCGTCCCTCGCTCATTTCCGGTGCGCCGTTTTTACATTTGCAATT CGAATCGGATTGAAATCGGAATCGGAgTCAGAATCGGCGCGAGGACA TTTCCATTCCTTCCCATTCTCTTTTTATTTTTTTTTTTTGCGATCTGCG TTGCATATGCGAGGGCTAATTAGCATGCGGCATTTCCAGCAATCAGA GTAGAGCGGCACAATAGGAATAATAACGCGGAATGGAAGAGGATGT GTATGCCTCGCTAGGTGCCTACAACGACAGCGGTGGCGACGATTGGA GCAGCTCGGAGCACCTGGTCCTGTGGGAGGAGGATGAGACGCAGCG

AACGACTGCTAATGCCACCAGTCGGCATAATCAACTGCATGTGGC CAGGTGGAATGCCACCGGCAATGCGACCATCAGCGCGACCTTCGAGG ACGTACCCTTCGATGCGAACAACTACTGGGCC

Toll-2,6,8 dsRNA was prepared as described in Lavalou, Mao et al. (ref. [37]).

## RNAi and Dextran injection in *Drosophila* embryos

For dsRNA injection experiments, embryos were collected within 1 h of egg laying, dechorionated in bleach, rinsed and aligned on coverslips coated with heptane glue. Embryos were desiccated for 5–6 min and covered with Halocarbon 200 oil and injected with RNAse-free water or dsRNA (5 µM concentration). Post injected embryos were stored at 18 °C prior to imaging.

Dextran 568 (5 mg/ml; 10,000 MW; Invitrogen), Dextran 488 (5 mg/ml; 70,000 MW; ThermoFisher) were injected on the perivitel-line space at the end of cellularization.

## *Drosophila* live imaging

Embryos were prepared as previously described[84], stage six embryos were collected from flies caged at room temperature (except for overexpression experiments, where flies were caged at 18 °C to allow sufficient maternal GAL4 deposition). Time-lapse images were acquired every 15 s for 30–45 min, on Nikon dual camera Eclipse Ti inverted spinning disk microscope (distributed by Roper) with a X-100/1.4 oil-immersion objective at ~21–22 °C, controlled by the Meta-morph software. In all, 11 z-stacks separated by 0.5 µm were acquired starting from the most apical plane. 491-nm and 561-nm lasers were used to excite GFP or mNeonGreen and mCherry fluorophores, respectively. Laser powers were measured and kept constant between controls and perturbed experiments. Bright field time-lapse images were acquired using an inverted microscope (Zeiss) and a program-mable motorized stage to record multiple positions over time (Mark&Find module from Zeiss), running on AxioVision software (Zeiss). Images were acquired every 1 min for 3–4 h.

## *Drosophila* image processing and data analysis

All image analyses were performed using Fiji freeware. All images were pre-processed with a 0.5 mean filter to smooth the background, fol-lowed by an average or maximum projection of the z-stacks (3–11 z-slices spaced by 0.5 µm, depending on the experiment). Background subtraction was performed by taking a rolling sphere of 50 pixels (~4 µm). For all quantifications of MyoII levels, an additional cyto-plasmic signal was subtracted by taking the mean intensity of the ROI of 10 pixels in diameter inside the cells from the 2D projected images, which excludes the apical signal. Ecad::GFP or Lifeact Cherry or GAP43::mCherry (depending on the experiment) were used to seg-ment the cells. For Rho1 sensor in 5HT2A loss of function and gain of function (Fig. 4b–h) and Rho1/MyoII in Toll triple RNAi knockdown (Supplementary Fig. 10), the Rho1 sensor signal was used to segment the cells. For the 5HT2A and Cirl genetic interaction (Fig. 3e–g), the average projection of the MyoII signal were used for the segmentation as well as tracking the centroid distance to quantify local extension. The resulting skeletons were dilated by 2 pixels on either side of the one-pixel wide cell-cell interface (5 pixels wide), excluding the vertices and used as junctional masks to extract junctional intensities. For the apical signal quantification, the individual cell masks were shrunk by 4 pixels to exclude the junctional signal using the macro by G.Kale[50]. Junctional and medial Rho1 or MyoII values were mean intensities calculated from the above masks. All the data points obtained were plotted using GraphPad Prism 9 software. All the images were seg-mented using the ImageJ Tissue Analyzer plugin (Aigouy et al., avail-able at https://github.com/baigouy/tissue_analyzer)[85].

To quantify the amplitude of polarity, the junctions obtained by the above method were categorized into different orientations, (0–15)° were considered as transverse junctions (AP junctions) and

(75–90)° as vertical junctions (DV junctions). Mean values of all junctions in the categories were measured and the amplitude of polarity was quantified by taking the ratio of the mean intensity of vertical and transverse junctions to ensure the robustness of the measurement[39,50,54].

To quantify the 5HT2A::mNeonGreen membrane signal (Supplementary Fig. 3e, f), the GAP43::mCherry signal was used to segment the cells. For the quantification of the 5HT2B::mCherry signal (Supplementary Fig. 7h, i), the average projection of the 5HT2B::mCherry signal was used for the segmentation. Mask obtained as described above were used to extract the mean values of the signals from the 2D images obtained by average projection of apical z-stacks corresponding to 7 μm. Number of vesicles (in Supplementary Fig. 8f″) were counted manually and normalized with the number of cells (30–60 cells per embryo) counted.

To quantify the co-localization of dextran with 5HT2A::mNeonGreen (Supplementary Fig. 8a′) or 5HT2B::mCherry (Supplementary Fig. 8b′), and 5HT2A::mNeonGreen and 5HT2B::mCherry (Supplementary Fig. 8c′), pixel-pixel intensities were correlated using custom-written Python code. To quantify the F-actin levels, Lifeact::mCherry signal were used to segment the cells and mean intensities were measured from the 2D projected images as described above.

**Tissue scale dynamics.** The extent of elongation in (Supplementary Fig. 1b, d) was measured by tracking the contact point between the perivitelline membrane and the posterior tissue near the pole cells (marked with pink * in Supplementary Fig. 1a, c) and normalized to the maximum length the tissue could traverse by measuring the distance between the posterior end of the embryo and the cephalic furrow. Local tissue elongation (Supplementary Fig. 2a, b, i) was measured by tracking (using the manual tracking Fiji plugin) the centroid of cells (on X-100 videos, frames acquired every 15 s, that were two cells apart for 15–20 min after the end of ventral pulling. Relative length was calculated as (L-L0)/L0, where L is the centroid distance between the two tracked cells at each time point and L0 is the initial centroid distance. The tracking data were plotted in Python. T1 events and rosettes were detected using the stable T1 tracker and the rosettes tracker, respectively, included in the ImageJ Tissue Analyzer plugin (Aigouy, Benoît et al., available at https://github.com/baigouy/tissue_analyzer)[85]. The tracker automatically detects cells that irreversibly lose contact. Rosettes were detected with the threshold of a minimum distance of ten pixels and contain minimum of five cells. The number of stable T1s or rosettes obtained at each time point was normalized to the number of cells tracked at that time point (on X60 videos, images acquired every 30 s or 1 min for 50 min after the cellularization front passed the nucleus) and plotted as cumulative events over 30 min (T0 is the onset of ventral pulling) in custom written Python code.

## Scanning fluorescence cross-correlation spectroscopy (sFCCS) in *Drosophila* embryo

**Sample preparation.** Embryos were prepared as described above in the *Drosophila* live imaging section. sFCCS measurements were performed during the slow phase of the germ-band extension on late stage 8 or early stage 9 embryos at room temperature -21–22 °C.

**Microscope setup.** sFCCS was performed on a Zeiss LSM 880 system (Carl Zeiss, Oberkochen, Germany) using a Plan-Apochromat 100x, 1.4NA oil-immersion objective. Samples were excited with a 488 nm Argon and a 561 nm diode laser in line interleaved excitation to minimize signal cross-talk. To split excitation and emission light, a 488/561 dichroic mirror was used. Fluorescence was detected between 491 nm and 754 nm on a 32 channel GaAsP array detector operating in photon counting mode. A pinhole of the size of an airy unit was used to reduce out-of-focus light. All measurements were performed at room temperature.

**Data acquisition.** For sFCCS, a line scan of 128 × 1 pixels (pixel size 63 nm) was performed perpendicular to the plasma membrane at the cell-cell interface with 403.20 μs scan time. Typically, 100,000 lines were acquired for each excitation line (total scan time ca. 80 s), alternating the two different excitation wavelengths. The effective time resolution is thus 806.4 μs, sufficient to reliably detect the diffusion dynamics observed in the samples described in this work (i.e., diffusion times ~10–100 ms). Laser powers were adjusted to keep photobleaching below 50% at maximum in both channels (average signal decays were ca. 40% for mNeonGreen and 20% for mCherry). Typical excitation powers were ca. 8 μW (488 nm) and ca. 12 μW (561 nm). Scanning data were exported as TIFF files, imported and analyzed in MATLAB (The MathWorks, Natick, MA, USA; version R2020a) using custom-written code.

**Data analysis.** sFCCS analysis was performed as previously described[62,86] with minor modifications. Briefly, all scan lines in each channel were aligned as kymographs and divided into blocks of 2000 lines. In each block, the lines were summed column by column and the lateral position of maximum fluorescence was determined. This position defines the membrane position in each block and was used to align all lines to a common origin. A fit of a function consisting of a Gaussian and a sigmoid component was then fitted to the fluorescence profile in each block. The sigmoid part of the fit was subtracted from all lines in the block, effectively removing background fluorescence. Finally, all aligned and background corrected line scans were temporally averaged over the entire kymograph and fitted with a Gaussian function. The obtained waist of this function was typically in the range of 0.25–0.5 μm, indicating that the corrections for lateral movement and background signal were effective. The pixels corresponding to the plasma membrane were defined as pixels within ± 2.5 SD of the Gaussian peak. In each line and channel, these pixels were integrated, providing the membrane fluorescence time series $F_i(t)$ in channel $i$. In order to correct for depletion due to photobleaching, a two-component exponential function was fitted to the fluorescence time series for each spectral species, $F_i(t)$, and a correction formula was applied[60,87]. Finally, the autocorrelation functions (ACFs; $g$= green channel, $r$ =red channel) and the cross-correlation function (CCF) were calculated as follows, using a multiple tau algorithm:

$$G_{auto,i}(\tau) = \frac{\langle \delta F_i(t)\delta F_i(t+\tau)\rangle}{\langle F_i(t)\rangle^2},$$

$$G_{cross}(\tau) = \frac{\langle \delta F_g(t)\delta F_r(t+\tau)\rangle}{\langle F_g(t)\rangle\langle F_r(t)\rangle},$$

where $\delta F_i(t) = F_i(t) - \langle F_i(t)\rangle$ and $i=g,r$.

To avoid artefacts caused by long-term instabilities or single bright events, CFs were calculated segment-wise (10 segments) and then averaged. Segments showing clear distortions (typically less than 20% of all segments) were manually removed from the analysis[60].

A model for two-dimensional diffusion in the membrane and Gaussian focal volume geometry[86] was fitted to all CFs:

$$G(\tau) = \frac{1}{N}\left(1 + \frac{\tau}{\tau_d}\right)^{-1/2}\left(1 + \frac{\tau}{\tau_d S^2}\right)^{-1/2}$$

To ensure convergence of the fit for all samples (i.e., ACFs and CCFs of correlated and uncorrelated data), positive initial fit values for the particle number $N$ and thus $G(\tau)$ were used. In the case of uncorrelated data, i.e., for CFs fluctuating around zero, this constraint can generate low, but positive correlation amplitudes due to noise. From the diffusion time $\tau_d$, the diffusion coefficient $D$ was determined by

$D = \frac{\omega_0^2}{4\tau_d}$. The waist $\omega_0$ was determined from point FCS measurements with AlexaFluor® 488 (Thermo Fisher Scientific, Waltham, MA, USA) dissolved in water at 20 nM, which were performed at the same laser power and 2 μm depth to minimize aberrations. The structure parameter $S$ was fixed to the average value determined in calibration measurements, typically around 5. From the determined particle number $N$, the protein surface concentration $c$ was quantified: $c = \frac{N}{A_{eff}} = \frac{N}{\pi\omega_0^2 S}$, where $A_{eff} = \pi\omega_0^2 S$ is the effective detection area[86].

Relative cross-correlation values (rel.cc.) were calculated from the amplitudes of the ACFs and CCFs:

$$rel.cc. = \max\left\{\frac{G_{cross}(0)}{G_{auto,g}(0)}, \frac{G_{cross}(0)}{G_{auto,r}(0)}\right\},$$

where $G_{cross}(0)$ is the amplitude of the CCF and $G_{auto,i}(0)$ is the amplitude of the ACF in the $i$-th channel. Whereas a rel.cc. value of 1 is theoretically expected for complete binding in 1:1 stoichiometry, experimentally determined rel.cc. values are usually lower due to limited overlap of the detection volumes of the two channels and non-fluorescent states of the fluorescent proteins used to tag the proteins of interest[61,88]. Thus, rel.cc. values obtained in positive control samples (e.g., fluorescent protein heterodimer constructs) are typically in the range of 0.4 to 0.7, depending on the choice of fluorophores[61,63,88].

## Chick experiments

**Chick embryo culture and time-lapse microscopy.** Chick embryos were collected at stage XI and cultured using a modified version of the EC culture system for 5–6 h prior to immunofluorescence processing or up to 15 h for live imaging experiments. Briefly, embryos were collected using paper filter rings and cultured on a semi-solid albumin/agarose nutrient substrate (mixture of albumin, agarose (0.2%), glucose and NaCl) at 37 °C in a humid chamber with or without drugs: DMSO for control embryos and Ritanserin at 20 μM, 50 μM, 100 μM, and 200 μM for treated embryos. For live imaging, embryos collected on filter paper rings were transferred to a bottom glass Petri dish (Mattek inc.) on the same media described above and imaged at 37 °C using an inverted microscope (Zeiss Apotome) with a ×5 objective.

**Chick immunofluorescence.** For antibody staining, chick embryos were fixed in ice-cold 4% formaldehyde/PBS for at least 1 h, permeabilized in PBS/0.1% Triton X-100 (PBT 0.1%), and blocked in PBT 0.1%/10% goat serum (from Gibco). Primary antibodies used in this study were mouse anti-ZO1 (Invitrogen ZO1-1A12) at 1:250 dilution and rabbit anti-pMyosin light chain 2 (Cell Signaling Technology CST-3671S and CST-3674S) at 1:50 dilution. Secondary antibodies conjugated to AlexaFluor 488 or 555 were purchased from ThermoFisher Scientific (Goat anti-Rabbit IgG (H + L) superclonal™ Secondary Antibody, AlexaFluor 488, Catalog Number A27034 and Goat anti-Mouse IgG (H + L) Superclonal™ Secondary Antibody, AlexaFluor 555, Catalog Number A28180) and used at 1:500 dilutions. Embryos were then mounted between slide and coverslip using Vectashield (Vector Laboratories) containing DAPI.

**Chick image acquisition.** Optical sections of fixed samples were obtained on a confocal microscope (Zeiss LSM 880 or LSM 700) using ×20 (Plan-Apochromat NA 0.8) objectives and ZEN software (Zeiss). Fiji software was used for image processing and data analysis.

**Analysis of tissue flows in chick embryo.** Following a common approach to describe the morphogenesis of epithelial sheets[89], the motion of the planar epiblast was described as a continuous, two-dimensional flow field. Time-lapse movies of embryos were analyzed using custom Java software based on the ImageJ API for image processing[35]. Briefly, particle image velocimetry (PIV) was used to evaluate the local displacement of the tissue between successive movie frames. The resulting displacement fields were used to reconstruct cell trajectories.

**Chick image analysis.** ZO1, which marks the tight junctions, stained images were used to segment the cells. Junction and cell masks were obtained using Tissue Analyzer plugin (Aigouy et al.[85]). The corresponding junctional phospho-MyoII intensities were measured from the above masks by importing into ImageJ ROI manager. To quantify the medial MyoII levels, cell masks were shrunk by 3 pixels to exclude junctional signal. MyoII polarity was quantified by taking the ratio of the mean intensities corresponding to the junctions forming supracellular cables (cables that span more than 2 cells) to that of the mean intensities of the junctions spanning single cell orthogonal to the supracellular cables. Junctions were selected manually and intensities were calculated manually by drawing ROIs corresponding to the selected junctions. All the data points obtained were plotted using GraphPad Prism 9 software.

## Statistics and reproducibility

*Drosophila* data were pooled from at least 4–37 independent experiments. Each embryo was considered an independent experiment. For double-stranded RNA injection experiments (Fig. 5a–d, f–h), each mount was considered as an independent experiment. All experiments were repeated at least three times. All overexpression and knockdown genetic crosses were repeated at least three times. Chick experiments were performed at least three times per condition on at least three embryos; each embryo was considered an independent experiment. In all experiments, we used appropriate controls, taking into account genetic backgrounds (please see 'Methods'), growth temperature, Myosin-II copy numbers (please see 'Methods'), and pharmacological treatments. All $P$ values were calculated using a two-tailed Mann–Whitney test in GraphPad Prism 9. In all figures, n.s.: $P > 0.05$, $*P \le 0.05$, $**P \le 0.005$, $***P \le 0.0005$, $****P \le 0.00005$. Box plots extend from the 25th to the 75th percentile and whiskers are minimum and maximum values. No statistical methods were used to determine sample size. The experiments were not randomized and the investigators were not blinded to allocation during the experiments and outcome assessment.

### Reporting summary

Further information on research design is available in the Nature Portfolio Reporting Summary linked to this article.

## Data availability

The data supporting the findings of this study and material are available on request from the corresponding author (T.L.). Source data are provided with this paper.

## Code availability

The codes used for image analysis, sFCCS data analysis, and chick PIV analysis are available upon request from the corresponding author (T.L.). The code used for sFCCS data analysis is also available at https://github.com/ValDunsing/ScanningFCCS.

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

## Acknowledgements

This work benefited greatly from the stimulating discussions in the Lecuit team and we thank all the lab members for their support during the course of the project. We are grateful to Leslie B. Vosshall (Rockefeller University, USA), Yi Rao (Peking University, China), David Krantz Lab (UCLA, USA), and the Drosophila Genetic Resource Center and the Bloomington Stock Center for the gift of flies. We thank Cedric Maurange lab for providing lab space for chick experiments. We also thank Pierre-françois Lenne for access to infrastructure and lab space. The authors thank Salvatore Chiantia (University of Potsdam, Germany) for technical discussions about the sFCCS analysis. This work was supported by the ERC AdvGrant Biomecamorph (323027), the Ligue Nationale Contre le Cancer (Equipe labellisée 2018). S.Ka. was initially supported by a Ph.D. fellowship from the LabEx INFORM (ANR-11- LABX-0054) and of the A*MIDEX project (ANR-11-IDEX-0001-02), funded by the "Investissements d'Avenir French Government program" and from Ligue Nationale Contre le Cancer, *4ème année de thèse*, (TDQD22642). V.D. acknowledges support by HFSP long-term postdoctoral fellowship (HFSP LT0058/2022-L). We acknowledge the France-BioImaging infrastructure supported by the French National Research Agency (ANR–10–INBS- 04-01, Investments for the future.

## Author contributions

S.Ka. and T.L. conceived the project. S.Ka. performed all the experiments and quantifications except for sFCCS experiments presented in Fig. 5j–o performed by S.Ka. and V.D., and quantified by V.D., and chick experiments performed and quantified by S.Ka. and M.S. (Fig. 6 and Supplementary Fig. 12). S.Ke. performed preliminary experiments and analysis of 5HT2A and 5HT2B mutants and 5HT2A overexpression during the onset of the project. E.D.S. and J.-M.P. created all the fluorescent constructs and dsRNA. C.M. contributed reagents for chick embryo experiments. S.Ka., V.D., M.S., and T.L. discussed the data. S.Ka. and T.L. wrote the manuscript and all authors made comments.

## Competing interests

The authors declare no competing interests.
