## [Peer Review File · Nature Communications]

Serotonin signaling regulates actomyosin contractility during morphogenesis in evolutionarily divergent lineagesREVIEWER COMMENTS

Reviewer #1 (Remarks to the Author):

The authors investigated the contribution of serotonin signaling via 5HT2A/2B to the morphogenesis of germ band extension in *Drosophila*. Cell rearrangement via the T1 process is one of the major mechanisms of epithelial tissue deformation and is closely related to the polarity of actomyosin activity localization. They found that serotonin signaling forms a signal module that quantitatively controls the amplitude of planar polarized MyoII contractility specified by Toll receptors and GPCR Cirl. Since serotonin receptors are ubiquitous in various organs, this mechanism may also be used in the morphogenesis of other animals and organs. In fact, the authors demonstrate its involvement in tissue flow and actomyosin activity control during the gastrulation period of chicken embryos. Overall, the novelty of the results is high, and I believe they will be of broad interest to the readers of the journal. However, there are concerns regarding the reproducibility of the data, and it is necessary to respond to the following comments before its acceptance.

Major comments:

1. In Fig.1e/i, junctional MyoII intensity in 5HT2A loss/gain-of-function was examined. What is the reason for the different values of control in each figure? For example, the values of control are quite different between Fig1e (top) and Fig1i (top). It would be helpful for readers if you could explain whether this difference is due to the normalization step when calculating the intensity, due to just fluctuations between experiments, or another reason.
2. In Fig.1f/j, MyoII distribution in 5HT2A loss/gain-of-function analysis was quantified. In Fig. 1f, at $T>0$, the majority of data points for both control and 5HT2A^{-/-} were distributed in the 2-3 range, and there was no statistically significant difference between the two. On the other hand, Fig. 1j concludes that there is a statistically significant difference between the control and 5HT2A^{+/+} at $T=10/15$. However, compared to Fig1f, the value of control at $T=10/15$ appears to be lower in Fig1j. If they were distributed in the range of 2-3 as in Fig1f, there would not be a significant difference. In other words, there may be no significant difference between control, 5HT2A^{-/-}, and 5HT2A^{+/+}. Since the amplitude of polarity is defined as the ratio of intensity (between DV and AP), I think that it is not affected by the variation of absolute signal intensity among experiments or methods for signal normalization. The number of data should be increased sufficiently to ensure reproducibility of the data distribution. This is important because it is closely related to the conclusion that hyper-polarization is really occurring in 5HT2A^{+/+}?
3. L119. More explanations are needed. How is the effect of decrease in T1 frequency complemented by an increase in rosettes formation during the tissue elongation process of 5HT2A^{+/+} embryos; is the shape of cells forming rosettes different from the rest? That is, do the contributions of cell rearrangement and cell shape changes to tissue-level (or cell population level) deformation differ between control and 5HT2A^{+/+} embryos?
4. In Fig2e-h, the MyoII distribution in Trh gain-of-function analysis was quantified. The median of MyoII amplitude of polarity in the control embryos in Fig2g is less than 2, while that in Fig1f is about 2.5. What could be the cause of this difference, attributed to the variation in values among the subpopulations? If the control embryos in Fig2g took a polarity amplitude of ~ 2.5 , they would not differ significantly from the Trh^{+/+} population, which would greatly affect the conclusion that ligand overexpression causes hyper-polarization.
5. In Fig4b-h, Rho1 activity in 5HT2A loss/gain of function analysis was quantified. In Fig. 4c (top), the time dependence of signal intensity in control embryos is small and almost constant at around 40-50, whereas in Fig. 4g (top), the signal intensity in control embryos appears to decrease with time. What is the cause of this qualitative difference? If it did not decay with time, there would not be a

significant difference in the signal intensity compared with 5HT2A++ embryos. This affects the interpretation of the results.

6. In Fig. 6, the roles of serotonin signaling in the regulation of tissue flow and myosin activity during chick gastrulation were examined. The analyses in *Drosophila* showed that the signaling via serotonin receptors does not affect the polarity of MyoII localization but is involved in the regulation of Junctional MyoII signaling intensity. Is the same true in the chick gastrulation system? Quantification of polarity and junctional MyoII levels by image analysis could answer this question.

Minor comments:

- L89. It might be helpful to have a little explanation in the main text of how rosettes are defined and how they are counted.
- Why is there no statistical test in Fig3c (bottom)?

Reviewer #2 (Remarks to the Author):

Karki et al introduce serotonin signalling to the control of polarised Myosin II contraction during early embryogenesis. Linking systemic hormones to cellular mechanism is exciting and extremely welcome, the hope being that this and future studies of this kind will demystify currently opaque physiological processes.

The authors do a very good job of setting serotonin receptor signalling within the context of other GPCR signalling and what is known about the control of junction contraction during axis extension and the paper is well organised, making it easy to read. I found the sequence of genetic/biochemical perturbations through the paper impressive, with various tools I had not come across before (so at times beyond my ability to evaluate). Unravelling the role of 5HT2B in regulating 5HT2A turnover was particularly impressive and no doubt took rather more work than the succinct section in the paper suggests. The evolutionary perspective, comparing fly with chick, was interesting and largely plausible (even though N=2 with possibilities of evolutionary loss and gain of serotonin signalling function in between).

Overall, this paper represents a large body of interesting work that clearly merits publication. I have what amount to minor criticisms.

1. The "amplitude of Myo-II contractility" was assessed as the ratio of AP to DV junctional Myosin concentrations, which is a plausible place to start but gets complicated when levels of myosin are low. Given the importance of this Myosin polarity measure throughout the paper (it is in almost every figure), I would have liked to have seen an explicit rationale for the use of this measure early in the paper [first used line 99]. An alternative would be to use the difference between AP and DV junctional concentrations, which would have the effect presumably of making differences less significant where overall concentrations are low, as in various perturbations in the paper. I don't particularly favour one or the other because I don't have evidence either way, but I thought the authors should try to present some evidence in favour of their adopted method. A claim based purely on theory that the ratio is the appropriate measure wouldn't convince me, since that would inevitably be based on various assumptions. Instead, this would mean showing that the rate of intercalation is proportional to the AP:DV ratio across a range of mean concentrations, and that this was a better predictor of intercalation rate than the AP-DV difference. I don't mean doing new experiments to show this, but pulling together existing data, given the wide range of perturbations under- and over-activating Myosin that the authors employ in this and previous papers from the group.

2. There is no mention of parasegment boundaries or Tartan/Ten-m, as revealed by e.g. Zallen and Sanson groups. PSBs seem relevant in two ways. First, do the multi-junction cables leading to rosettes

in the hyper-polarised germbands resemble the way PSBs are arranged and how they intercalate? So vertical lines of junctions between PSBs, controlled by toll-2/6/8, become PSB-like? Second, are PSBs changed by the various serotonin signalling perturbations in a different way to the between-PSB toll-2/6/7-controlled junctions? I'd recommend some explicit mention of what is the same or different about the effect of serotonin signalling on PSBs and non-PSB vertical junctions in the paper.

Small things:

Line 95: what is "dorsolateral cell division"? Are these the non-neural ectoderm divisions that happen at the same time as the mesectoderm divisions? If so, 20 minutes before that is in the middle of the fast phase of GBE, is that what is meant? Pls clarify.

Section starting line 145 on serotonin itself: Including the word 'autocrine' somewhere near the start of this section would help a lot, if that is indeed how serotonin is thought to signal in the germband, and would stop the reader wondering. Is there any literature that helps with over what range is 5HT thought to diffuse extracellularly? Is serotonin signalling also autocrine in the chick primitive streak?

Response to REVIEWERS' COMMENTS

We would like to thank the reviewers for taking time to review the manuscript. We appreciate their feedback and helpful comments.

Here are our point-by-point responses to the reviewers' comments:

Reviewer #1 (Remarks to the Author):

The authors investigated the contribution of serotonin signaling via 5HT2A/2B to the morphogenesis of germ band extension in *Drosophila*. Cell rearrangement via the T1 process is one of the major mechanisms of epithelial tissue deformation and is closely related to the polarity of actomyosin activity localization. They found that serotonin signaling forms a signal module that quantitatively controls the amplitude of planar polarized MyoII contractility specified by Toll receptors and GPCR Cirl. Since serotonin receptors are ubiquitous in various organs, this mechanism may also be used in the morphogenesis of other animals and organs. In fact, the authors demonstrate its involvement in tissue flow and actomyosin activity control during the gastrulation period of chicken embryos. Overall, the novelty of the results is high, and I believe they will be of broad interest to the readers of the journal. However, there are concerns regarding the reproducibility of the data, and it is necessary to respond to the following comments before its acceptance.

→ Thank you for highlighting the significance of our work. We have responded to your comments point by point below.

Major comments:

1. In Fig. 1e/i, junctional MyoII intensity in 5HT2A loss/gain-of-function was examined. What is the reason for the different values of control in each figure? For example, the values of control are quite different between Fig 1e (top) and Fig 1i (top). It would be helpful for readers if you could explain whether this difference is due to the normalization step when calculating the intensity, due to just fluctuations between experiments, or another reason.

→ Thanks for pointing this out and for giving us an opportunity to clarify something we omitted to comment on in the original submission. The difference in controls is due to the use of different controls. To monitor MyoII we used a construct that expresses the regulatory light chain of MyoII, Sqh, fused to mCherry, under the sqh promoter. However, depending on experiments, we could use 1 or 2 copies of this transgene and this is the reason why we had to use different controls (1 or 2 copies of sqh-Sqh::mCherry, referred to as Sqh::mCherry) and why indeed values may be different. The 5HT2A loss-of-function was done with two copies of Sqh::mCherry (Fig 1e), while the gain-of-function was done with a single copy of Sqh::mCherry (Fig 1i). For 5HT2A overexpression, we used the UAS/GAL4 system. For early 5HT2A overexpression in the embryo, the 67-GAL4/UAS-5HT2A must be maternally loaded into the embryo. Since all transgenes required for 5HT2A overexpression and visualization (UAS-5HT2A, 67-GAL4, sqh-sqh::mCherry and Ecad::GFP for cell contour marking) are located on chromosome 2, their proximity makes recombination of the four transgenes impossible. We therefore crossed heterozygous ;67GAL4, sqh-sqh::mCherry, Ecad::GFP /UAS-5HT2A; F1-female progenies with homozygous UAS-5HT2A males to

express 5HT2A both maternally and zygotically. Half of the embryos imaged have only a single copy of all transgenes: 67GAL4, sqh-sqh::mCherry, UAS-5HT2A and Ecad::GFP. So, the MyoII levels are indeed higher in two-copies controls (Fig1e) compared to the single-copy controls (Fig 1i).

We have now explicitly commented on the use of single or two copies of the sqh-Sqh::mCherry transgene, MyoII levels in different controls, and mentioned the genetics in the methods section under the subsection “Fly strains and genetics” and now reads: “Homozygous *Ecad::GFP,sqh-sqh::mCherry* line was used to visualize MyoII in 5HT2A loss-of-function (Fig. 1b-g), Trh loss-of-function (Fig. 2b-d), 5HT2B loss-of-function (Extended Data Fig. 6a-d), 5HT2B knock-down (Fig. 5a-d) and 5HT2A and 5HT2B double knock-down (Fig. 5f-h) and therefore have two copies of *sqh-sqh::mCherry*. All the over-expression experiments to visualize MyoII have a single copy of *sqh-sqh::mCherry* or *sqh-sqh::GFP* and were performed using the *67GAL4,Ecad::GFP; sqh-sqh::mCherry* line (ref. 39) except for Fig. 3h-k where the *67GAL4;sqh-sqh::GFP,Lifeact::mCherry* line was used. MyoII levels are higher in the two copy controls compared to the single copy controls. All the genetic crosses are detailed below.”

2. In Fig.1f/j, MyoII distribution in 5HT2A loss/gain-of-function analysis was quantified. In Fig. 1f, at $T > 0$, the majority of data points for both control and 5HT2A^{-/-} were distributed in the 2-3 range, and there was no statistically significant difference between the two. On the other hand, Fig. 1j concludes that there is a statistically significant difference between the control and 5HT2A^{+/+} at $T = 10/15$. However, compared to Fig1f, the value of control at $T = 10/15$ appears to be lower in Fig1j. If they were distributed in the range of 2-3 as in Fig1f, there would not be a significant difference. In other words, there may be no significant difference between control, 5HT2A^{-/-}, and 5HT2A^{+/+}. Since the amplitude of polarity is defined as the ratio of intensity (between DV and AP), I think that it is not affected by the variation of absolute signal intensity among experiments or methods for signal normalization. The number of data should be increased sufficiently to ensure reproducibility of the data distribution. This is important because it is closely related to the conclusion that hyper-polarization is really occurring in 5HT2A^{+/+}?

→ Thanks again for pointing this out. These differences in the amplitude of polarity in the controls in Fig. 1f (5HT2A loss-of-function) and Fig. 1j (5HT2A gain-of-function) are observed due to the different copy numbers of MyoII and the use of slightly different controls which were each best appropriate for the experiment conducted. In the 5HT2A loss-of-function, we have two copies, so MyoII levels are higher, while in the 5HT2A gain-of-function, there is a single copy of MyoII. Also, in 5HT2A^{+/+}, we have maternally loaded 67-GAL4 to drive the over-expression of the UAS-5HT2A, so we have compared with appropriate control containing 67-GAL4 in the background. Hence, the controls in these two conditions (5HT2A-loss and 5HT2A gain-of-function), do not have same genetic background on top of having different copy numbers of Sqh::mCherry. As can be seen in Fig 1e and 1i, the levels of MyoII in DV and AP junctions do not exactly double in two copies compared to the single copy controls so the polarity index is not perfectly identical. The amplitude of polarity is slightly higher in two copies than in a single copy (however, the mean values are within a range of 2-2.5 in both controls, while in 5HT2A^{+/+} it is around 3). This is clearly illustrated in the image in Fig.1h. Because of the genetics involved, we weren't able to do these experiments in the same genetic background, so we had to use appropriate controls.

3. L119. More explanations are needed. How is the effect of decrease in T1 frequency complemented by an increase in rosettes formation during the tissue elongation process of 5HT2A⁺⁺ embryos; is the shape of cells forming rosettes different from the rest? That is, do the contributions of cell rearrangement and cell shape changes to tissue-level (or cell population level) deformation differ between control and 5HT2A⁺⁺ embryos?

→ We observe a decrease in T1 frequency in 5HT2A⁺⁺ (Fig.1 m), and the decrease in T1s is expected to result in a decrease in tissue extension. However, tissue extension is not affected (Extended Data Fig. 2i). Since both the T1s and rosettes contribute to tissue extension (Ref. 31,48), we expected an increase in rosettes frequencies, as indeed observed (Extended Data Fig. 2j,k). While in control, T1s appear first, followed by rosettes (References: Blankenship, J Todd et al. ; Tamada, Masako et al.; Paré, Adam C et al.) (after ~10min; Fig.1m and Extended Data Fig.2k), in 5HT2A⁺⁺, rosettes appear 5 minutes earlier than in controls (Extended Data Fig.2k). Thus, the effect of the decrease in T1s in 5HT2A⁺⁺ is compensated for by the increase in rosette formation to contribute the same amount of tissue extension.

The increased and earlier formation of rosettes in 5HT2A⁺⁺ embryos is consistent with increased polarization of MyoII and formation of longer supracellular cables as seen in Fig. 1h. Indeed, the Zallen group reported that rosettes require the formation of supracellular cables of actomyosin: contraction of such longer cables draw more than 4 cells together in a high-order vertex. This is also expected to reduce the number of T1s as in T1s 4 cells form a 4-way vertex by shrinkage of a single contact between neighbouring cells.

We have added the following sentences in the main text in L122: “While, in control, T1s appear first, followed by rosettes (after 10min) (Fig. 1m and Extended Data Fig. 2k) (36,48). In 5HT2A gain-of-function, rosettes appear 5 minutes earlier.”

References:

Blankenship, J Todd et al. “Multicellular rosette formation links planar cell polarity to tissue morphogenesis.” *Developmental cell* vol. 11,4 (2006): 459-70. doi:10.1016/j.devcel.2006.09.007

Tamada, Masako et al. “Abl regulates planar polarized junctional dynamics through β -catenin tyrosine phosphorylation.” *Developmental cell* vol. 22,2 (2012): 309-19. doi:10.1016/j.devcel.2011.12.025

Paré, Adam C et al. “A positional Toll receptor code directs convergent extension in *Drosophila*.” *Nature* vol. 515,7528 (2014): 523-7. doi:10.1038/nature13953

4. In Fig2e-h, the MyoII distribution in Trh gain-of-function analysis was quantified. The median of MyoII amplitude of polarity in the control embryos in Fig2g is less than 2, while that in Fig1f is about 2.5. What could be the cause of this difference, attributed to the variation in values among the subpopulations? If the control embryos in Fig2g took a polarity amplitude of ~2.5, they would not differ significantly from the Trh⁺⁺ population, which would greatly affect the conclusion that ligand overexpression causes hyper-polarization.

→ Thanks again for this, these differences are due to different copy numbers of MyoII::mCherry as explained under points 1 and 2 above. In Fig1f the amplitude of polarity is calculated in a condition with two copies of MyoII, while in Fig2g with a single copy of MyoII and there is also maternally loaded 67-GAL4 in the background. The amplitude of polarity is slightly higher with two copies than with a single copy. We think that these two subpopulations are not comparable for the same reason described earlier in comment-2. We have mentioned the genetics in the methods section. We used appropriate controls each time.

5. In Fig4b-h, Rho1 activity in 5HT2A loss/gain of function analysis was quantified. In Fig. 4c (top), the time dependence of signal intensity in control embryos is small and almost constant at around 40-50, whereas in Fig. 4g (top), the signal intensity in control embryos appears to decrease with time. What is the cause of this qualitative difference? If it did not decay with time, there would not be a significant difference in the signal intensity compared with 5HT2A⁺⁺ embryos. This affects the interpretation of the results.

→ The qualitative difference could result from a different level of expression of the sensor due to position effect of transgenesis and the temperature (see below). It is well known that there is a position effect on the expression level of the same construct depending on which attP landing site transgenesis occurred.

The two genotypes used differ in the chromosomal insertion of the Rho1 sensor construct "UBI-mEGFP::AniLRBD":

- for the loss-of-function experiment, sensor is inserted in 2L
- and
- for gain-of-function, sensor is inserted in 3L

Another difference could be due to the growth temperature of the two cages: loss-of-function cage was incubated at room temperature (22°C) and gain-of-function cage at 18°C to allow sufficient maternal deposition of 67-GAL4 to drive the expression of UAS 5HT2A (lower temperature increases expression from the maternal 67-GAL4 line).

We have explicitly mentioned the growth temperature in the Methods under *Drosophila* live imaging subsection.

Moreover, in acknowledgement of your comments-1,2,4, and 5 above; we have now explicitly mentioned the statement of experimental conditions in the Statistics and Reproducibility section and reads: "In all experiments, we used appropriate controls, taking into account genetic backgrounds, growth temperature, MyoII copy numbers, and pharmacological treatments."

6. In Fig. 6, the roles of serotonin signaling in the regulation of tissue flow and myosin activity during chick gastrulation were examined. The analyses in *Drosophila* showed that the signaling via serotonin receptors does not affect the polarity of MyoII localization but is involved in the regulation of Junctional MyoII signaling intensity. Is the same true in the chick gastrulation system? Quantification of polarity and junctional MyoII levels by image analysis could answer this question.

→ Yes, serotonin signaling in chick embryos affects the levels of MyoII only, not the planar polarity. Junctional MyoII levels can be seen reduced but polarized as shown in magnified images in the Fig. 6b and Extended Data Fig. 12a, and quantified in Extended Data Fig. 12b,d.

We have now added the following sentences in the main text in L391: “Both the junctional and medial phospho-MyoII levels were reduced (Fig. 6b and Extended Data Fig. 12a-c) while the amplitude of polarity was not affected in the treated embryos (Extended Data Fig. 12a-d). This suggests that 5HT2A/2B signaling is required for MyoII activation and regulation of MyoII levels but not polarization during chick gastrulation, consistent with that in *Drosophila*.”

We discussed this further in the discussion of chick data in L434 and reads: “We found that the serotonin/receptors signaling also activates MyoII contractility and regulates MyoII quantity but not polarity during chick gastrulation.”

Chick image analysis is described in the Methods under “Chick experiments”, *chick image analysis* subsection.

Minor comments:

- L89. It might be helpful to have a little explanation in the main text of how rosettes are defined and how they are counted.

→ We used ImageJ Tissue Analyzer plugin (Aigouy, Benoît et al., available at https://github.com/baigouy/tissue_analyzer) (Ref. 88) to detect and track rosettes. We counted the rosettes with minimum 5 cells. We have explained it in the methods section.

Reference: Aigouy, Benoît et al. “Cell flow reorients the axis of planar polarity in the wing epithelium of *Drosophila*.” *Cell* vol. 142,5 (2010): 773-86. doi:10.1016/j.cell.2010.07.042

- Why is there no statistical test in Fig3c (bottom)?

→ Thank you for pointing this out, we have added it.

Reviewer #2 (Remarks to the Author):

Karki et al introduce serotonin signalling to the control of polarised Myosin II contraction during early embryogenesis. Linking systemic hormones to cellular mechanism is exciting and extremely welcome, the hope being that this and future studies of this kind will demystify currently opaque physiological processes.

The authors do a very good job of setting serotonin receptor signalling within the context of other GPCR signalling and what is known about the control of junction contraction during axis extension and the paper is well organised, making it easy to read. I found the sequence of genetic/biochemical perturbations through the paper impressive, with various tools I had not come across before (so at times beyond my ability to evaluate). Unravelling the role of 5HT2B in regulating 5HT2A turnover was particularly impressive and no doubt took rather more work than the succinct section in the paper suggests. The evolutionary perspective, comparing fly with chick, was interesting and largely plausible (even though $N=2$ with possibilities of evolutionary loss and gain of serotonin signalling function in between).

Overall, this paper represents a large body of interesting work that clearly merits publication. I have what amount to minor criticisms.

→ Thank you for your appreciation of the importance of our work. We believe, we have done our best to investigate the role of serotonin/receptors signaling using the contemporary tools and techniques. We welcome your comments and are happy to address them below.

1. The “amplitude of Myo-II contractility” was assessed as the ratio of AP to DV junctional Myosin concentrations, which is a plausible place to start but gets complicated when levels of myosin are low. Given the importance of this Myosin polarity measure throughout the paper (it is in almost every figure), I would have liked to have seen an explicit rationale for the use of this measure early in the paper [first used line 99]. An alternative would be to use the difference between AP and DV junctional concentrations, which would have the effect presumably of making differences less significant where overall concentrations are low, as in various perturbations in the paper. I don't particularly favour one or the other because I don't have evidence either way, but I thought the authors should try to present some evidence in favour of their adopted method. A claim based purely on theory that the ratio is the appropriate measure wouldn't convince me, since that would inevitably be based on various assumptions. Instead, this would mean showing that the rate of intercalation is proportional to the AP:DV ratio across a range of mean concentrations, and that this was a better predictor of intercalation rate than the AP-DV difference. I don't mean doing new experiments to show this, but pulling together existing data, given the wide range of perturbations under- and over-activating Myosin that the authors employ in this and previous papers from the group.

→ Thank you for this comment. The absolute values of the MyoII intensities could fluctuate within an experimental condition (i.e. genotype) due to potential variations in imaging conditions. Although we take great care having stable imaging conditions (e.g., laser power, or state of the CCD camera) we need to have measurements that are robust to possible fluctuations in imaging conditions in a given experiment, and as we repeat it another day. Since we are interested in the asymmetric enrichment of the junctional MyoII, the ratio is more robust. Indeed, if the signal is multiplied by a certain factor alpha due to whatever fluctuation in imaging, in the ratio analysis DV/AP , polarity is the same, but if we measure the difference

DV-AP, the polarity would be multiplied by alpha. The ratio analysis is in itself of form of normalization though it may underestimate the real polarity if the signal to noise ratio is lower. We, and others have adopted this analysis of polarity early on (such as in References, 51, Kale, Girish R et al.; 39, Garcia De Las Bayonas, Alain et al.; 64, Munjal, Akankshi et al.) and indeed the ratio measurement of polarity is robust.

We have now mentioned it in the Methods under *Drosophila* image processing and data analysis and cited the following references.

References:

Kale, Girish R et al. "Distinct contributions of tensile and shear stress on E-cadherin levels during morphogenesis." *Nature communications* vol. 9,1 5021. 27 Nov. 2018, doi:10.1038/s41467-018-07448-8

Garcia De Las Bayonas, Alain et al. "Distinct RhoGEFs Activate Apical and Junctional Contractility under Control of G Proteins during Epithelial Morphogenesis." *Current biology : CB* vol. 29,20 (2019): 3370-3385.e7. doi:10.1016/j.cub.2019.08.017

Munjal, Akankshi et al. "A self-organized biomechanical network drives shape changes during tissue morphogenesis." *Nature* vol. 524,7565 (2015): 351-5. doi:10.1038/nature14603

2. There is no mention of parasegment boundaries or Tartan/Ten-m, as revealed by e.g. Zallen and Sanson groups. PSBs seem relevant in two ways. First, do the multi-junction cables leading to rosettes in the hyper-polarised germbands resemble the way PSBs are arranged and how they intercalate? So vertical lines of junctions between PSBs, controlled by toll-2/6/8, become PSB-like? Second, are PSBs changed by the various serotonin signalling perturbations in a different way to the between-PSB toll-2/6/7-controlled junctions? I'd recommend some explicit mention of what is the same or different about the effect of serotonin signalling on PSBs and non-PSB vertical junctions in the paper.

→ Thanks for raising this interesting question. Indeed, the non-PSB vertical junctions are indistinguishable from the PSB vertical junctions in both the 5HT2A and serotonin gain-of-function cases where longer cables form along with increased polarization. This can be seen in Fig.1h, Extended Data Fig. 2h and Video 3 (5HT2A gain-of-function); Fig. 2e and Video 5 (Trh gain-of-function).

In the case of 5HT2A loss-of-function, there is an overall reduction in MyoII levels in both the PSB and non-PSB vertical junctions (Fig.1d and Video 2), but PSB are still visible. Trh loss-of-function does not affect junctional MyoII levels and PSB are visible (Video 4). In 5HT2B loss-of-function there is an overall increase in all the junctional MyoII levels (Video 14), PSB are quite difficult to distinguish from non-PSB. Since, we cannot really draw an overall conclusion about the effect of serotonin/receptor signaling on PSB vs non-PSB vertical junctions we have not mentioned it explicitly in the manuscript, however highlighted it in case of the hyper-polarization with orange arrowheads as in Fig. 1h, Fig. 2e,i,l and Extended Data Fig. 2h.

We have mentioned in the case of hyperpolarization following 5HT2A overexpression in L117 and now reads: “In line with this, we observed more aligned DV-oriented junctions forming supracellular cables that are indistinguishable from the parasegment boundaries in 5HT2A overexpressing embryos (Fig. 1h and Extended Data Fig. 2h)”

Small things:

Line 95: what is “dorsolateral cell division”? Are these the non-neural ectoderm divisions that happen at the same time as the mesectoderm divisions? If so, 20 minutes before that is in the middle of the fast phase of GBE, is that what is meant? Pls clarify.

→ These are cell divisions in the dorsal ectoderm, described in Collinet et.al (reference 46). Thus, “20 minutes before that” refers to when the tissue starts to move posteriorly. It is definitely in the fast phase of GBE, but not sure if it is precisely the middle (more or less close to the middle).

We have made changes in the main text L97 as “the cell divisions in the dorsal ectoderm” and cited Collinet et.al (reference 46).

Section starting line 145 on serotonin itself: Including the word ‘autocrine’ somewhere near the start of this section would help a lot, if that is indeed how serotonin is thought to signal in the germband, and would stop the reader wondering. Is there any literature that helps with over what range is 5HT thought to diffuse extracellularly? Is serotonin signalling also autocrine in the chick primitive streak?

→ Indeed, we don’t know if Serotonin is autocrine or paracrine. Since it is a small diffusible molecule, in this context, we would assume that it can elicit signaling in the neighboring cells as well. But, the diffusivity of ligand could be affected by the presence of receptors at the cell surface. We haven’t come across any literature that reports the diffusion coefficient of serotonin in extracellular space. We would like to access 5HT diffusion in the embryo with FCS, but we lack proper tools to visualize the dynamics. We assume it is likely the same in the chick primitive streak.

REVIEWERS' COMMENTS

Reviewer #1 (Remarks to the Author):

The authors have responded appropriately to all my concerns.

Reviewer #2 (Remarks to the Author):

Thank you for your responses that adequately address my concerns.

For the Myosin II polarity ratio measure, I agree that normalising the variation between embryos of the same genotype is a high priority, and trust that even with overall low Myosin intensities the measure remains robust given how widely this measure has already been used.

Thank you for the clarification of what happens to PSBs vs non-PSB intensities in your different perturbations. It looks likely that 5HT signalling affects all junctions in a similar manner, preserving differences so far as you can tell, so the question about whether it affects Trn differently from Toll-2/6/8 seems somewhat moot.